# Beta activity in human anterior cingulate cortex mediates reward biases

Jiayang Xiao[1,2], Joshua A. Adkinson[1], John Myers[1], Anusha B. Allawala[3], Raissa K. Mathura[1], Victoria Pirtle[1], Ricardo Najera[1], Nicole R. Provenza [1], Eleonora Bartoli[1], Andrew J. Watrous[1], Denise Oswalt [1], Ron Gadot[1], Adrish Anand [1], Ben Shofty [4], Sanjay J. Mathew [5], Wayne K. Goodman[5], Nader Pouratian [6], Xaq Pitkow[2,7,8], Kelly R. Bijanki [1], Benjamin Hayden [1] & Sameer A. Sheth [1,2,5,7] ✉

The rewards that we get from our choices and actions can have a major influence on our future behavior. Understanding how reward biasing of behavior is implemented in the brain is important for many reasons, including the fact that diminution in reward biasing is a hallmark of clinical depression. We hypothesized that reward biasing is mediated by the anterior cingulate cortex (ACC), a cortical hub region associated with the integration of reward and executive control and with the etiology of depression. To test this hypothesis, we recorded neural activity during a biased judgment task in patients undergoing intracranial monitoring for either epilepsy or major depressive disorder. We found that beta (12–30 Hz) oscillations in the ACC predicted both associated reward and the size of the choice bias, and also tracked reward receipt, thereby predicting bias on future trials. We found reduced magnitude of bias in depressed patients, in whom the beta-specific effects were correspondingly reduced. Our findings suggest that ACC beta oscillations may orchestrate the learning of reward information to guide adaptive choice, and, more broadly, suggest a potential biomarker for anhedonia and point to future development of interventions to enhance reward impact for therapeutic benefit.

One of the hallmarks of human behavior is the ability to associate rewards with the stimuli that produce them[1–3]. The anticipation of a reward can influence our decision-making process by overriding our initial sensory judgments and subsequently altering our subjective evaluation of a stimulus[4–7]. These alterations in judgment, though sometimes suboptimal, can also be beneficial by biasing our behavior in a manner that increases the likelihood of obtaining greater rewards[8]. A fundamental unresolved question in human neuroscience is,

therefore, what neural processes drive the learning, or association of reward information, so as to bias subsequent decisions[9,10].

Dysfunction of these reward association processes is a common characteristic of several neuropsychiatric disorders, most notably major depressive disorder[11–15]. Depression is a disabling disease characterized by anhedonia, disengagement, and reduced enjoyment of life[16,17]. Among its hallmarks is the diminished influence of reward anticipation on decision-making[18,19]. Investigating the underlying

[1]Department of Neurosurgery, Baylor College of Medicine, Houston, TX 77030, USA. [2]Department of Neuroscience, Baylor College of Medicine, Houston, TX 77030, USA. [3]School of Engineering, Brown University, Providence, RI 02912, USA. [4]Department of Neurosurgery, University of Utah, Salt Lake City, UT 84112, USA. [5]Department of Psychiatry and Behavioral Sciences, Baylor College of Medicine, Houston, TX 77030, USA. [6]Department of Neurological Surgery, UT Southwestern Medical Center, Dallas, TX 75390, USA. [7]Department of Electrical and Computer Engineering, Rice University, Houston, TX 77005, USA. [8]Center for Neuroscience and Artificial Intelligence, Baylor College of Medicine, Houston, TX 77030, USA. ✉e-mail: sasheth@bcm.edu

mechanisms of this phenomenon is therefore important for the development of innovative interventions aimed at providing therapeutic benefit by enhancing the impact of reward[20–22].

Several human electrophysiological studies have identified a mediofrontal oscillatory component associated with positive feedback in both gambling task and reversal learning task, tasks that have key features in common with our bias task. The increase observed in these tasks is in the beta range and occurs 200 to 400 ms after the feedback informing the participant about the monetary gains[23–27]. It has been proposed that this beta activity is generated in the prefrontal cortex; the most commonly inferred source site is the anterior cingulate cortex (ACC). It is further assumed that the ACC then transmits a fast motivational value signal, still in the beta band, from the frontal cortex to downstream reward-related regions[26]. Moreover, beta activity in the cingulate cortex is of particular importance in depression[28–30]. For example, a recent study showed that beta activity best tracks depressive states, seen as a decrease in beta band power during the first month of chronic stimulation, followed by an eventual rise[31]. This result suggests that sustained, antidepressant responses might involve increased beta band power after prolonged stimulation.

Existing evidence suggests that associative learning processes occur, in part, within specialized reward and control circuitry[3,32]. In particular, the ACC is a notable hub region that is associated with both reward and executive functions, and that has been linked to their integration[33,34]. Crucially, extensive research highlights the importance of the ACC in representing reward information and monitoring rewarding outcomes, as well as in facilitating learning and enabling strategic adjustments[35–39]. Moreover, altered activity in the ACC and regions connected to it has been associated with depression[40–46].

Anhedonia, the loss of interest and pleasure from normally rewarding stimuli, is a cardinal symptom of depression and is often inadequately treated by traditional antidepressants[21,47]. While individuals diagnosed with depression may exhibit a range of symptoms, anhedonia stands out as one of two key symptoms required for a diagnosis of major depressive disorder[48]. Moreover, it is prevalent in other psychiatric and neurological disorders including substance use disorder, bipolar disorder, schizophrenia, and Parkinson's disease[49–52]. To highlight the common dimensions underlying mental health disorders, the National Institute of Mental Health provides a research framework called the Research Domain Criteria (RDoC)[53]. Within this framework, the probabilistic reward task (PRT) is a well-validated task to objectively measure anhedonia. In this task, unequal frequency of reward between two correct responses produces a response bias towards the more frequently rewarded stimulus[5]. While prior studies have shown that performance predicts depression severity, the underlying electrophysiological basis of anhedonia remains unclear[6,53,54].

We recorded intracranial local field potentials (LFPs) from four reward-related regions in human participants performing the probabilistic reward task. In subjects with medically refractory epilepsy undergoing intracranial seizure monitoring ("Epilepsy Cohort", no clinical diagnosis of major depressive disorder), we find that enhanced beta (12–30 Hz) oscillations after decision choice in the ACC predict stronger biasing and also track reward receipt. On the other hand, in subjects with severe treatment-resistant depression ("Depression Cohort", i.e., a cohort undergoing intracranial monitoring as part of a clinical trial studying depression; NCT03437928), both the behavioral bias and the neural response towards rewarding stimulus are reduced. These results suggest that ACC beta oscillations may reflect neural processes that orchestrate the binding of reward and sensory information to guide adaptive choice. Additionally, our findings imply that these oscillations could serve as a potential biomarker for anhedonia, paving the way for future research to develop targeted neuromodulatory interventions.

## Results

### Participants developed response bias towards the more frequently rewarded stimulus

The Epilepsy Cohort consisted of 15 participants with medically refractory epilepsy, but no diagnosis of a mood disorder, undergoing intracranial seizure monitoring. These subjects performed a variant of the probabilistic reward task (Fig. 1a)[5]. Each run of the PRT consisted of three blocks of 100 trials. On each trial, following a fixation period, a mouthless cartoon face appeared (0.5 s). Then a mouth appeared (variable length; long mouth: 54 mm, short mouth: 49 mm) for 0.1 s and then disappeared. Participants were asked to identify whether the previously presented mouth was long or short. Participants received no reward for an incorrect response and probabilistically received either a reward (dollar sign) or no reward (empty rectangle of same size and shape) for a correct response. The likelihood of receiving a reward was fixed at either 60% or 20% depending on whether the choice was designated as the rich stimulus or the lean stimulus.

The optimal strategy, in terms of correct performance and reward accumulation, is to ignore the rich/lean status and choose based solely on perceptual features (i.e., mouth size). The behavioral measurement of this task was response bias, or bias towards choosing the rich stimulus. Here, following the conventions of Pizzagalli et al., 2005, we use the term response bias to indicate the extent to which behavior is modulated by reinforcement history.

Epilepsy Cohort participants showed a response bias: they tended to choose the more frequently rewarded stimulus more often than the less frequently rewarded one. Response bias averaged across all participants was larger than zero for all three blocks (Fig. 1b, one-sample $t$ test compared with zero, block1: $p = 0.036$, block2: $p = 0.0029$, block3: $p = 0.0061$). Consistently, the accuracy for the rich stimulus was higher than the accuracy for the lean stimulus (Fig. 1c, paired-sample $t$ test, block1: $p = 0.026$, block2: $p = 0.0034$, block3: $p = 0.0086$). To assess the impact of feedback from the previous trial, we compared choice patterns following reward and no-reward outcomes (these analyses include all choices, not just correct ones, Fig. 1d). When there was a reward in the previous trial, the likelihood of choosing the same response (meaning long vs. short) was significantly larger (paired-sample $t$ test, $t = 5.9$, $p < 10^{-4}$), supporting the idea that the participants have modified their behavior based on the previous feedback.

### Rich stimulus induces larger ACC beta power during delay period

We recorded from four regions in the Epilepsy Cohort: anterior cingulate cortex (ACC), medial orbitofrontal cortex (mOFC), lateral orbitofrontal cortex (lOFC), and amygdala (Fig. 1e, Supplementary Table 1).

We first analyzed the delay period (Fig. 2a), which extends 500 ms after choice. The neural activity during this period occurs after the decision-making process. Cognitive effort typically occurs during the decision-making process, as individuals engage in mental processes such as evaluating options, weighing consequences, and selecting an action. Therefore, we do not think that the difference in neural activity between different options during the delay period, after the participant has already made the choice, should be strongly influenced by cognitive effort. During this period, sensory information and reward history can be integrated to generate value representations for stimuli, enabling the brain to anticipate the potential rewards and potentially promote learning[55,56]. We computed the average spectral power at six major frequencies (delta, theta, alpha, beta, gamma, and high-gamma) during the delay period (using the same definitions as in Xiao et al., 2023). Considering the observation of beta activity during reward processing and the importance of beta activity in depression, our core experimental hypothesis was in favor of the beta band. Results from the other five bands with Bonferroni correction were reported for reasons of completeness.

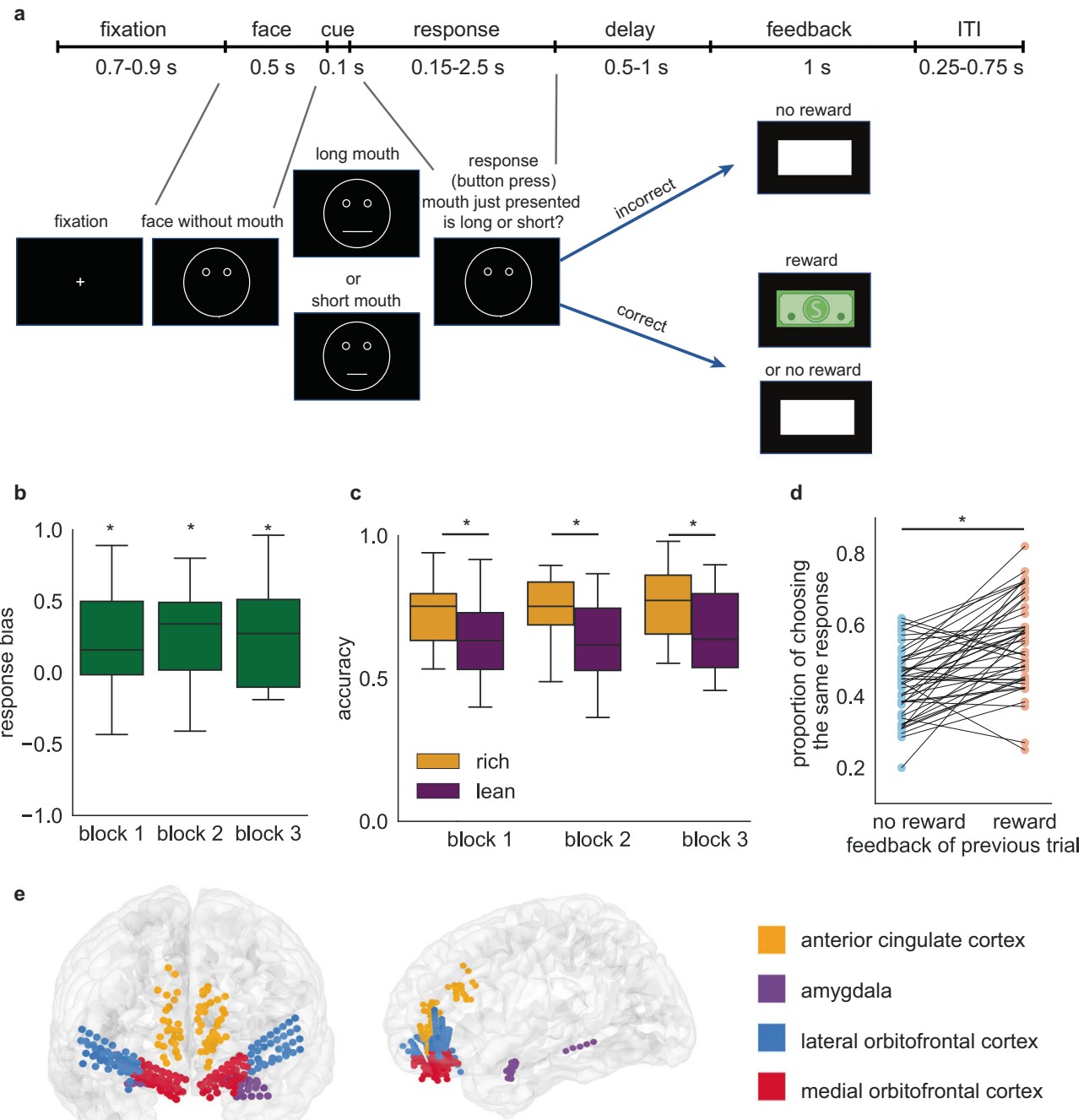

**Fig. 1 | Task description, behavioral performance, and recording locations.**
**a** Timeline of the probabilistic reward task. **b** Response bias averaged across all Epilepsy Cohort patients. A positive response bias value indicates there is a preference for choosing the more frequently rewarded stimulus. Boxplots illustrate quartiles at 25% and 75%, with horizontal lines denoting medians, and whiskers extending to 1.5 times the interquartile ranges. $n = 16$ runs of task. One-sample $t$ test compared with zero, block1: $p = 0.036$, block2: $p = 0.0029$, block3: $p = 0.0061$. '*' represents $p < 0.05$. **c** Accuracy for rich and lean stimuli averaged across all epilepsy patients. Boxplots illustrate quartiles at 25% and 75%, with horizontal lines denoting medians, and whiskers extending to 1.5 times the interquartile ranges. $n = 16$ runs of task. Paired-sample $t$ test, block1: $p = 0.026$, block2: $p = 0.0034$, block3: $p = 0.0086$. '*' represents $p < 0.05$. **d** Proportion of choosing the same response when there is a reward or no reward following the response in the previous trial. $n = 48$ blocks of task. Paired-sample $t$ test, $p = 3.9*10^{-7}$. '*' represents $p < 0.05$. **e** Intracranial recording electrodes sample reward-relevant regions in Epilepsy Cohort patients. Source data are provided as a Source Data file.

We found that beta power in ACC was greater for rich trials than for lean trials during the delay period (Fig. 2b, linear mixed model, t(stimulus) = 2.1, $p = 0.039$, coef = 2.5, 95% confidence interval (95% CI) = [0.12–4.8], analysis epoch: 0–500 ms after choice). This difference was limited to the beta frequency - the other five frequency bands did not show any detectable difference ($p > 0.05$ in all cases). Moreover, the majority of channels exhibited greater beta power in response to

the rich stimulus than the lean stimulus (Fig. 2b). Specifically, we found that event-aligned activity increases immediately after choice; this rise was significantly larger following choice of the rich stimulus (Fig. 2c, cluster-based permutation test, the significant cluster begins at 244 ms and ends at 293 ms after the choice, number of permutations = 1000).

Interestingly, beta modulation in the lateral orbitofrontal cortex was also significant before multiple comparison corrections (albeit in

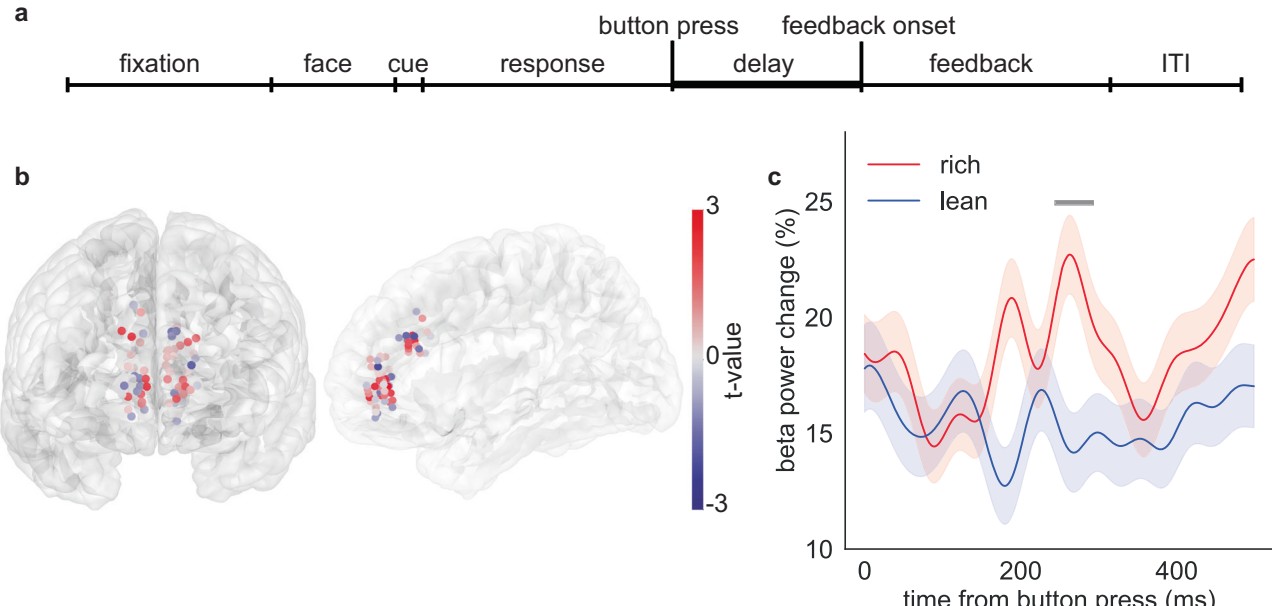

**Fig. 2 | Increase in ACC beta power in response to the rich stimulus during the delay period. a** Delay period starts from the button press and ends when feedback appears on the screen. **b** Differences in beta (12–30 Hz) power between the rich and lean trials are plotted at each sEEG contact. The color represents the *t* value comparing rich with lean trials. Red indicates greater power during rich trials. **c** Beta power change during the delay period. Red indicates rich trials while blue indicates lean trials. Trials are time-locked to the button press. The horizontal bar indicates time points in the cluster that are statistically significant (*p* < 0.05), as determined by two-sided cluster-based permutation test. Data are presented as mean values ± SEM. Source data are provided as a Source Data file.

the opposite direction, linear mixed model, t(stimulus) = −2.1, coef = −3.3, 95% confidence interval (95% CI) = [−6.4 − −0.17], *p* = 0.039). This observation is notable given the hypothesized role of OFC in reward learning processes, and given its close interconnections with ACC[32,57–60]. This result suggests that reward biasing may reflect the activity of a broader circuit consisting of ACC and lOFC, possibly working together.

On the other hand, these effects were not brain-wide. Specifically, the beta modulation in medial orbitofrontal cortex and amygdala was not significant (t(stimulus) = −1.7, *p* = 0.09, coef = −5.1, 95% confidence interval (95% CI) = [−12.8–2.7], for mOFC beta, t(stimulus) = −1.3, *p* = 0.20, coef = −2.1, 95% confidence interval (95% CI) = [−4.4–0.3], for amygdala beta, before multiple comparison corrections). Note that the failure to achieve significance is not itself dispositive; indeed, we report other evidence linking mOFC and amygdala in associations, suggesting that ACC may be part of a larger network focused on learning (see below).

## Neural activity during delay period is correlated with response bias

Next, we explored whether ACC beta activity plays a role in translating internal representations of stimulus value into behavioral preferences. Behavior preference is measured by response bias, with a stronger bias indicating a larger preference towards rewards. Therefore, a correlation with bias is the key indicator of learning the association between the reward and the stimulus.

During the delay period, neural activity distinguished rich and lean stimuli in blocks when bias was higher than the median, but not in blocks when it was lower than the median (Fig. 3a, b). In the high response bias condition, one significant cluster (cluster-based permutation test, from 234 ms to 322 ms, number of permutations = 1000) was found, while no significant cluster was found in the low response bias condition (Fig. 3b). These findings suggest that ACC activity covaries with the degree of bias. This pattern is illustrated in heatmaps showing the difference between the rich and lean stimuli at each contact (Fig. 3a): in the majority of channels, a notable increase in

power towards the rich stimulus was observed in the high response bias blocks. Conversely, the channels in the low response bias blocks exhibited fewer discernible differences between the rich and lean stimuli. Furthermore, we found that average beta power showed a significant difference between response to the rich and lean stimuli solely within the high response bias blocks (Fig. 3c, t(stimulus) = 3.2, *p* = 0.0013, coef = 5.4, 95% confidence interval (95% CI) = [2.1–8.7]), while no such distinction was observed within the low response bias blocks (t(stimulus) = −0.44, *p* = 0.66, coef = −0.76, 95% confidence interval (95% CI) = [−4.1–2.6]).

We then explored whether this neural activity during the delay period correlates with the behavioral variable, response bias. The difference in beta power response towards the rich or lean stimulus is positively correlated with response bias, but there is no significant correlation in other frequency bands (Fig. 3d, e, *r* = 0.38 for ACC beta, *p* = 0.013). The preference for choosing a more frequently rewarded stimulus increases as the difference in beta responses increases. This indicates that the ability to differentiate stimulus values, as reflected by ACC beta activity during the delay period, is associated with participants' behavior.

These results clearly implicate ACC beta activity in processing related to biasing. We found some evidence that OFC is likewise involved in reward learning. Response bias was positively correlated with the difference in beta power response towards rich or lean stimulus in the orbitofrontal cortex (*p* = 0.065 for mOFC beta, *p* = 0.037 for lOFC beta before multiple comparison corrections).

## Reward feedback elicits beta oscillations in the anterior cingulate cortex

Recognizing the crucial role of ACC beta power in reward anticipation during the delay period, next we tested whether this neural feature is also involved in the response to reward feedback. The observation of this feature during both the delay period and feedback period may suggest a common neural mechanism underlying the evaluation of stimuli and outcomes. The feedback period starts when either reward feedback or neutral feedback appears on the screen and ends when the

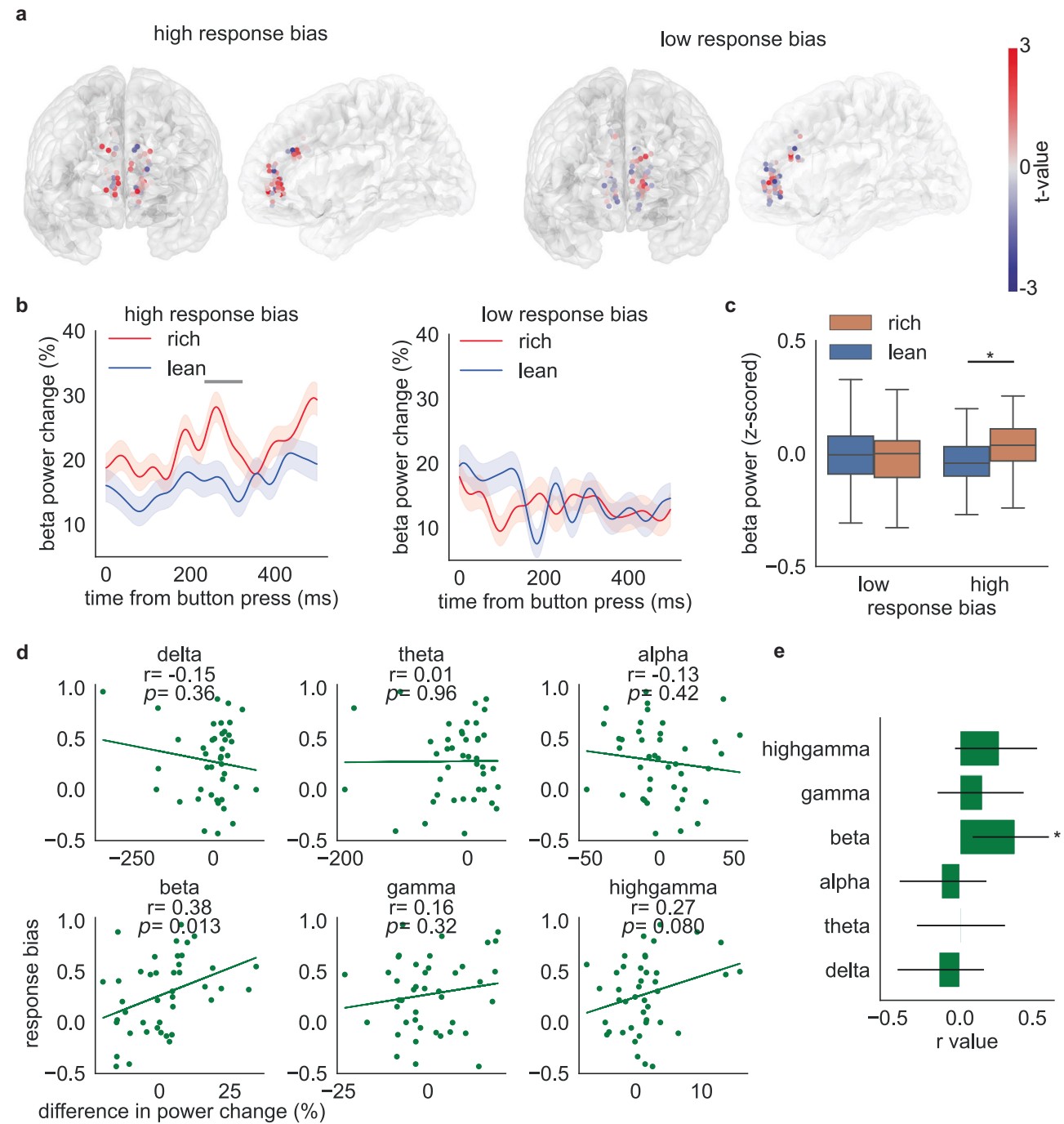

**Fig. 3 | Relationship between response bias and neural activity during the delay period. a** Differences in beta power between the rich and lean trials are plotted at each sEEG contact for high response bias blocks or low response bias blocks. **b** Beta power change during the delay period for high response bias blocks or low response bias blocks. Trials are time-locked to the button press. Horizontal bar depicts time points for which neural activity between rich trials and lean trials is significantly different (two-sided cluster-based permutation test, $p < 0.05$). Data are presented as mean values ± SEM. **c** Difference in beta power in high or low response bias blocks. Boxplots illustrate quartiles at 25% and 75%, with horizontal lines denoting medians, and whiskers extending to 1.5 times the interquartile ranges.

$n = 56$ channels for low response bias blocks and $n = 52$ channels for high response bias blocks. Linear mixed model, $p = 0.66$ for the low response bias blocks and $p = 0.0013$ for the high response bias blocks. '*' represents $p < 0.05$. **d** Relationship between response bias and the difference in reward response towards rich or lean stimulus in all frequency bands. The statistical test for the Pearson correlation coefficient is two-sided, and no adjustments were made for multiple comparisons. **e** Correlation between neural activity and response bias in all frequency bands. Center for the error bars represent the correlation coefficient. Error bars represent a 95% confidence interval. $p = 0.013$ for beta band. '*' represents $p < 0.05$. Source data are provided as a Source Data file.

image disappears (duration: 1000 ms) (Fig. 4a). We first computed the average spectral power at the beta frequency during the feedback period. To compare the activity between reward trials and neutral trials, we used a two-sample $t$ test to fit the spectral power for each channel in the ACC. Remarkably, most channels across the ACC exhibited greater beta power in response to reward feedback compared to neutral feedback (Fig. 4b). When trials were time-locked to the feedback onset, we observed a substantial increase in ACC beta

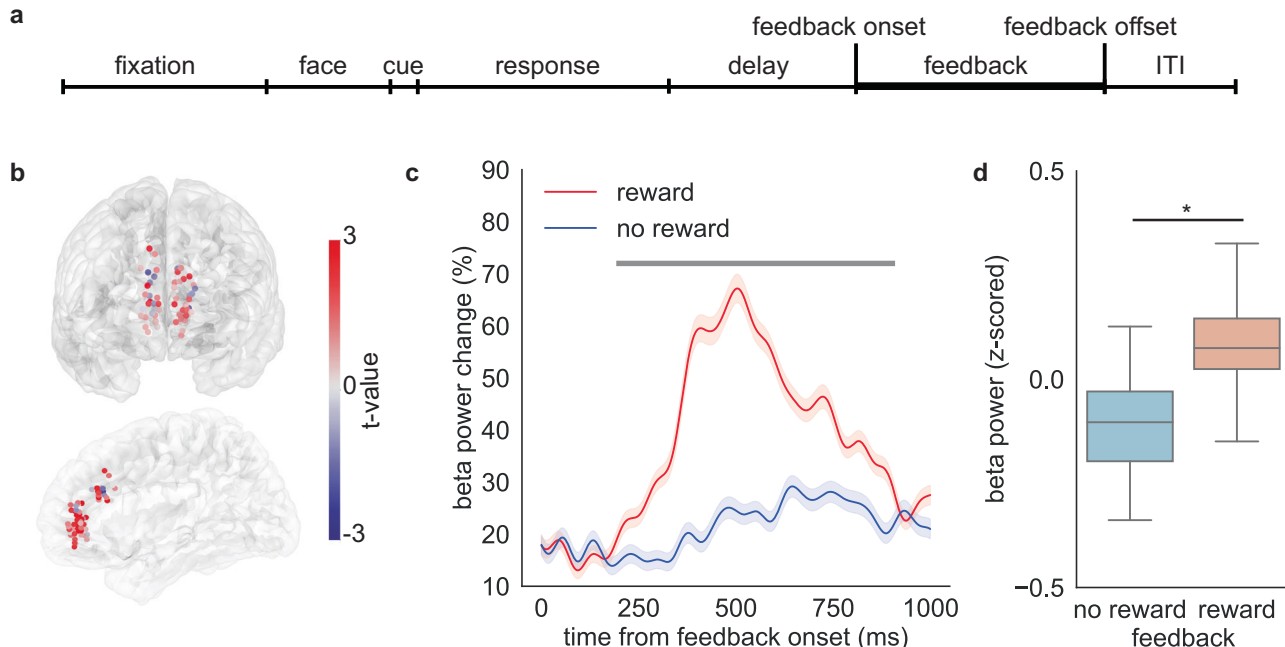

**Fig. 4 | Increase in ACC beta power in response to reward during the feedback period. a** The feedback period starts when the dollar bill or empty rectangle appears on the screen and ends when the feedback image disappears. **b** Differences in beta power between the reward and no-reward trials are plotted at each sEEG contact. The color represents the t value comparing reward feedback with neutral feedback. Red indicates greater power during the reward feedback. Only trials with correct response are included in the analysis. **c** Feedback-aligned beta power change averaged over trials and participants. Red indicates reward trials while blue indicates correct trials with no reward. Trials are time-locked to feedback onset. Horizontal bar depicts time points for which neural activity between reward trials and no-reward trials is significantly different (two-sided cluster-based permutation test, $p < 0.05$). Data are presented as mean values ± SEM. **d** Average beta power change across the feedback period. Boxplots illustrate quartiles at 25% and 75%, with horizontal lines denoting medians, and whiskers extending to 1.5 times the interquartile ranges. $n = 65$ channels. Linear mixed model, $p = 1.8*10^{-36}$. '*' represents $p < 0.05$. Source data are provided as a Source Data file.

power in response to reward (Fig. 4c, cluster-based permutation test, the significant cluster begins at 194 ms and ends at 907 ms after the feedback onset). This increase in the beta range reaches its peak approximately 500 ms after feedback onset.

To further confirm the effect of feedback type on the beta power change, we used a linear mixed effect model. The average beta power change during reward feedback was significantly larger than the change observed during neutral feedback (Fig. 4d, linear mixed effect model, t(feedback) = 12.6, $p < 10^{-4}$, coef = 15.7, 95% confidence interval (95% CI) = [13.3–18.2]). Overall, these results demonstrated that there was a beta power increase in response to reward in the ACC. This suggests that the ACC is engaged in the evaluation of both stimuli and outcomes, potentially representing a common neural mechanism underlying the assessment of reward values.

We found evidence that ACC is not alone in this process; indeed, two other regions from which we recorded, mOFC and amygdala, also showed a systematic relationship between post-reward beta activity and reward (mOFC: t(feedback) = −2.0, $p = 0.04$, coef = −3.4, 95% confidence interval (95% CI) = [−6.8 − −0.09], amygdala: t(feedback) = −5.0, $p < 10^{-4}$, coef = −9.8, 95% confidence interval (95% CI) = [−13.6 − −6.0], before multiple comparison corrections). It is interesting that amygdala showed a significant reversed relationship even after multiple comparison corrections; only ACC showed the same, positive, relationship during both the post-choice and delay periods.

To further confirm the result, we performed additional analysis using 12.5–30 Hz as the frequency range for beta activity[61]. As in the original result, we found that beta activity showed a discernible difference between the rich and lean stimuli within the high response bias blocks (Supplementary Fig. 1a), t(stimulus) = 3.1, $p = 0.0021$, coef = 5.1, 95% confidence interval (95% CI) = [1.8–8.3], while no such distinction within the low response bias blocks (t(stimulus) = −0.13, $p = 0.90$, coef = −0.22, 95% confidence interval (95% CI) = [−3.6–3.2]). During the

feedback period, beta activity during reward feedback was significantly larger than neutral feedback (Supplementary Fig. 1b, t(feedback) = 12.9, $p < 10^{-4}$, coef = 17.6, 95% confidence interval (95% CI) = [14.0–21.1]). We used time-frequency maps to investigate how various components of beta activity contribute to the overall effect (Supplementary Fig. 1c, d). Our findings indicate a large effect in the lower range of beta during both the delay and feedback periods, suggesting that our results are predominantly influenced by lower beta.

## Reward biasing and its neural correlates in ACC are blunted in depression cohort

In addition to the 15 patient Epilepsy Cohort, we also performed intracranial recordings in a group of four individuals with severe treatment-resistant depression ("Depression Cohort") participating in an NIH-funded clinical trial (NCT03437928, Fig. 5a)[62–65]. We analyzed behavioral performance and electrophysiological patterns in these depression patients. We did not observe any significant difference between the accuracy for the rich stimulus and the accuracy for the lean stimulus (Supplementary Fig. 2a, paired-sample $t$ test, block1: $p = 0.90$, block2: $p = 0.94$, block3: $p = 0.60$). Response bias in depression patients was not significantly different from zero for all three blocks (Supplementary Fig. 2b, one-sample $t$ test compared with zero, block1: $p = 0.93$, block2: $p = 0.83$ block3: $p = 0.38$). Compared to the epilepsy group, we found that response bias was blunted in these patients, especially in the last block of the task (Fig. 5b, two-sample $t$ test compared with epilepsy patients, $t = 2.3$, $p = 0.030$). This indicates that the behavioral preference towards more frequently rewarded stimuli was reduced in this severely depressed group of individuals. Then we investigated the neural response in ACC towards reward-related stimulus in this cohort. Unlike in the Epilepsy Cohort, there was no significant difference towards rich or lean stimulus in beta activity

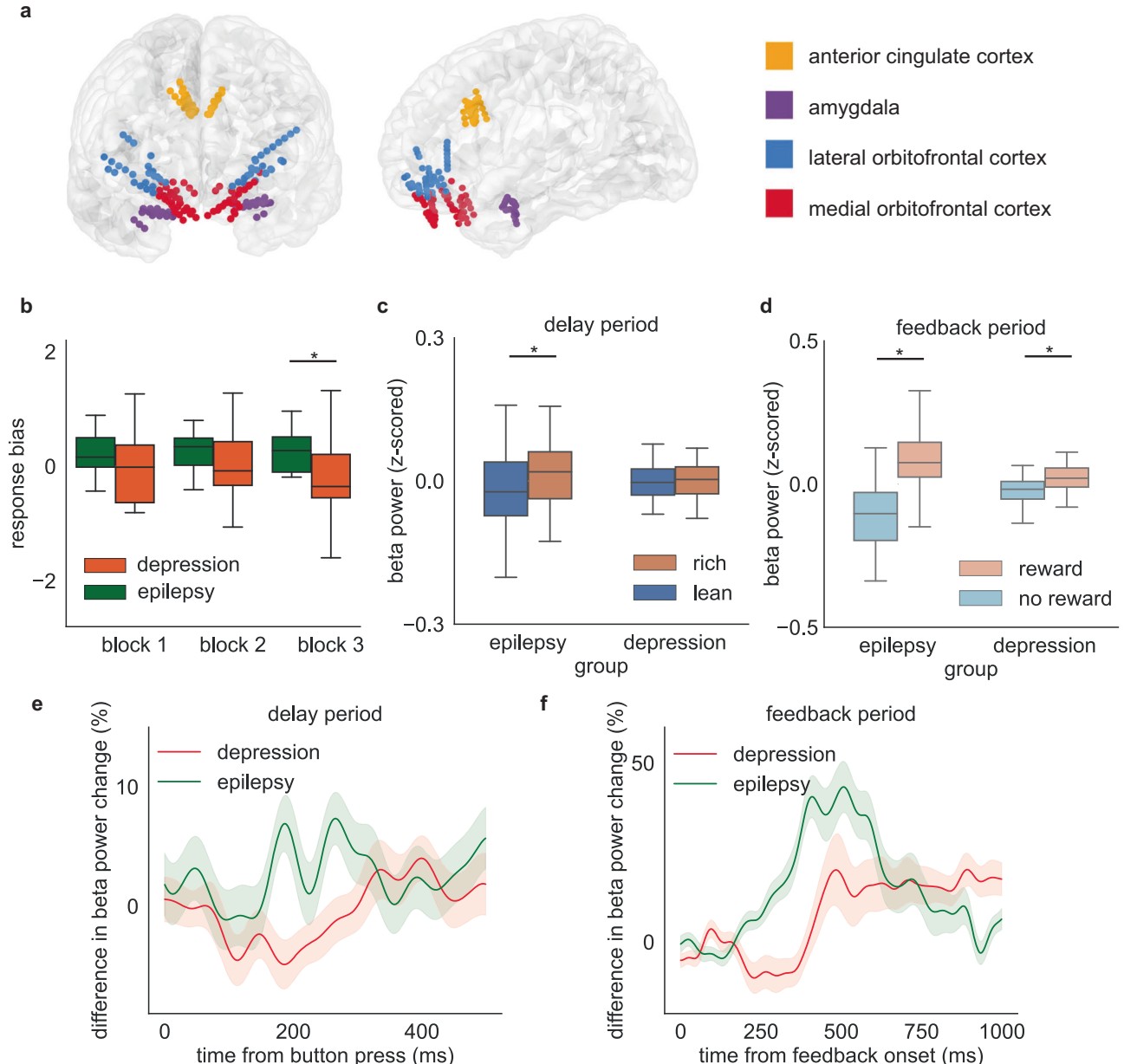

**Fig. 5 | Difference in reward response between Depression Cohort and Epilepsy Cohort. a** Intracranial recording electrodes sample reward-relevant regions in depression patients. **b** Response bias in Depression Cohort and Epilepsy Cohort. Boxplots illustrate quartiles at 25% and 75%, with horizontal lines denoting medians, and whiskers extending to 1.5 times the interquartile ranges. $n = 16$ runs for Epilepsy Cohort and $n = 12$ runs for Depression Cohort. Two-sample $t$ test, block1: $p = 0.24$, block2: $p = 0.24$, block3: $p = 0.030$. '*' represents $p < 0.05$. **c** Average beta power change across the delay period. Boxplots illustrate quartiles at 25% and 75%, with horizontal lines denoting medians, and whiskers extending to 1.5 times the inter-quartile ranges. $n = 65$ channels for Epilepsy Cohort and $n = 36$ channels for Depression Cohort. Linear mixed model, $p = 0.039$ for Epilepsy Cohort, $p = 0.77$ for

Depression Cohort. '*' represents $p < 0.05$. **d** Average beta power change across the feedback period. Boxplots illustrate quartiles at 25% and 75%, with horizontal lines denoting medians, and whiskers extending to 1.5 times the interquartile ranges. $n = 65$ channels for Epilepsy Cohort and $n = 36$ channels for Depression Cohort. Linear mixed model, $p = 1.8*10^{-36}$ for Epilepsy Cohort, $p = 1.6*10^{-11}$ for Depression Cohort. '*' represents $p < 0.05$. **e** Difference towards rich or lean stimulus in the beta activity across the delay period in Depression Cohort and Epilepsy Cohort. Data are presented as mean values ± SEM. **f** Difference towards reward or neutral feedback in the beta activity across the feedback period. Data are presented as mean values ± SEM. Source data are provided as a Source Data file.

during the delay period (Fig. 5c, linear mixed model, t(stimulus) = −0.29, $p = 0.77$, coef = −3.9, 95% confidence interval (95% CI) = [−3.0–2.2]). This result suggests a lack of reward-oriented anticipation in patients with severe depression. During the feedback period, beta power exhibited a larger change during reward feedback compared to that during neutral feedback (linear mixed model, t(feedback) = 6.7, $p < 10^{-4}$, coef = 7.5, 95% confidence interval (95% CI) = [5.3–9.7]). However, we observed that this beta power change was

significantly smaller in the Depression Cohort than it was in the Epilepsy Cohort (Fig. 5d, linear mixed model, t(feedback*group) = 4.7, $p < 10^{-4}$, coef = −8.2, 95% confidence interval (95% CI) = [−11.6 − −4.8]). When analyzing the temporal response, we observed that the difference towards rich or lean stimulus was diminished across the delay period in depression patients (Fig. 5e). When the trials were time-locked to the onset of feedback, we found that the reward response in depression patients was both reduced and delayed compared to the

Epilepsy Cohort (Fig. 5f). Apart from the original analysis using beta activity, we also reported the result in other frequency bands for both the Epilepsy Cohort and the Depression Cohort (Supplementary Table. 2). These findings imply alterations in reward processing within the ACC among individuals with depression.

## Discussion

We obtained multi-site intracranial recordings from patients undergoing inpatient monitoring for either epilepsy or treatment-resistant depression. Patients performed a perceptual discrimination task in which rewards were stochastic and biased. We found that enhanced beta oscillations in the ACC (and, less consistently, in regions connected to it), correlate with a stronger biasing effect and track the receipt of rewards. These results suggest that beta oscillations in the ACC may be a neural signature of processes associated with reward biasing, and especially the association of reward value with choice. Both the behavioral bias and the neural response to rewarding stimuli are diminished in patients with depression, highlighting the changes in reward processing within the ACC in depression. Previous studies have shown inconsistent results in behavioral findings during reinforcement learning in depression, possibly arising from the heterogeneity and stage of depression[5,6,21,54,66–69]. Given that increased anhedonia levels are associated with greater illness severity and longer episodes, the observed response bias difference in our study may be attributed to the advanced stage and high severity of treatment-resistant depression[70].

Anhedonia, with its profound impact on an individual's quality of life, can create challenges in various aspects including relationships, work, and daily functioning. Traditional antidepressants often fail to adequately address this symptom[21,47]. Thus, gaining a deeper understanding of anhedonia is important for improving diagnosis and treatment of depression. The perceptual bias task is the most commonly used task for the assessment of anhedonia, a major element of clinical depression[53]. There are two reasons for this: first, the task has direct face validity to the association of reward and action; second, there is ample empirical evidence linking changes in behavior in this task with anhedonia. It also has been used to assess reward learning and feedback responsiveness in both rodents and humans[71–74]. Our results, which include both correlations with the behavior in non-clinically-depressed individuals and reduction in reward response in individuals with severe depression, suggest that beta activity within the ACC may be a biomarker for anhedonia. Such a biomarker has many potential benefits, including the ability to improve diagnosis and symptom monitoring. Moreover, they present an appealing target for neuromodulatory trials, which could focus on altering ACC beta and thereby reducing anhedonia. In our study, the average beta power values were z-scored across trials, regardless of stimulus types within each contact. Therefore, the smaller values in depression patients indicate a lesser difference between conditions, not the absence of beta activity overall. Monitoring this neural feature in a more naturalistic environment is essential for comparison with healthy controls and crucial for the development of potential treatments. The limited enrollment of only four depression patients in our study is due to the unique clinical trial using an inpatient intracranial platform for therapy development. Future studies with larger sample sizes will be necessary to extend the generalizability of our findings to the broader population of individuals with depression. In comparison to traditional questionnaires, continuous and passive monitoring of this neural feature in patients requires less effort from them, offers greater objectivity, and facilitates timely intervention. Future work will need to determine the time course of changes in this potential biomarker relative to those of depressive symptoms.

While certain symptoms like tremor or stiffness in Parkinson's disease exhibit rapid moment-to-moment fluctuations, depressive symptoms are generally characterized by a more gradual and prolonged evolution over the course of days to months[75–77]. Therefore, this biomarker holds the potential to offer clinicians a valuable temporally dynamic signal about the individual's ongoing state and the transitions they experience. Such a signal could alert clinicians to be watchful and influence decisions regarding potential therapeutic maneuvers such as medication adjustments, behavioral interventions, or modifications in stimulation delivery. Further investigation is needed to assess its effectiveness and practical application. Current technology enables the monitoring of beta activity at the stimulation site in freely moving Parkinsonian patients, allowing for the precise control of stimulation delivery[78–80]. Similar to this approach, it is possible to track the instantaneous power of the beta band in depression patients by deep brain stimulation systems with sensing capabilities. While the intracranial signal provides more precise anatomical information via direct contact with brain tissue, future studies should aim to validate and replicate our observation of ACC beta activity using non-invasive approaches. Wireless EEG headsets may be necessary to achieve more frequent and convenient measurements, while the implementation of advanced source localization techniques can enhance anatomical precision. Advancements in these areas can facilitate broader application across patient populations.

These results also have implications for our understanding of the role of the ACC. The ACC is thought to play a crucial role in reward processing. In non-human primates, individual ACC neurons process both experienced and fictive rewards to dynamically guide changes in behavior. Monkeys with ACC lesions are impaired in using rewarded trials to sustain the selection of the correct object, emphasizing the importance of the ACC in reward-based decision-making[81]. Our study reveals that beta oscillations in the ACC, which represents reward outcome, are also elicited by rich stimuli and are correlated with behavioral preference. Previous electrical stimulation studies suggest that dACC plays a crucial role in motivation and drive[82,83]. In our study, we observed an increase in ACC activity when participants anticipated a reward. This anticipation of reward could potentially translate into an increased willingness to persevere and exert effort to obtain the reward. These stimulation studies also imply that adjacent brain regions may fulfill unique roles. Electrical stimulation of the anterior midcingulate cortex induces a 'will to persevere,' whereas stimulating subgenual or retrosplenial cingulate regions fails to evoke perceptual or behavioral responses. Further investigations with increased recording sites in the ACC are necessary to clarify the roles of its various subregions in reward processing. These findings provide further support for the importance of the ACC in reward learning. Indeed, these results are consistent with previous theories linking the ACC with cognition related to reward in general and to reward-mediated learning specifically[34,81,84,85]. In experiments involving both monetary gains and losses, research has demonstrated that an increase in theta power is associated with losses, while an increase in beta power is associated with gains[23,25,27]. Aligned with prior findings, our study showed a rise in beta activity within the ACC following positive feedback. Future studies could investigate ACC activity during a similar task that incorporates negative feedback for a more comprehensive understanding.

The alignment of reward information with other forms of sensorimotor processing involves the dynamic and flexible linking of inherently disparate pieces of information[26,86]. Several theories highlight the potential involvement of high-frequency oscillations in orchestrating learning and binding processes[87–92]. Our results indicate that successful reward biasing on sensorimotor decision-making is associated with specific enhancements in beta oscillatory activity within the ACC and areas in its local circuit. This finding suggests that ACC beta activity could play a key role in linking rewards to various cognitive areas. Future studies might explore how altering this activity could impact cognitive and emotional well-being.

## Methods

### Participants

Fifteen participants (eight males and seven females, mean age 39 years, range 19–60 years) undergoing invasive monitoring for the treatment of refractory epilepsy at Baylor St. Luke's Medical Center (Houston, Texas, USA) participated in our study. These participants did not carry a diagnosis of major depressive disorder. Implantation sites were determined solely by the clinical team for localization of the seizure onset zone. The Institution Review Board at Baylor College of Medicine approved this study (IRB protocol number H-18112), and all participants provided verbal and written consent to participate.

Four patients with treatment-resistant depression (two males and two females, mean age 42 years, range 37–58 years) who were enrolled in an early feasibility trial (NCT03437928) also participated in our study. This trial of individualized deep brain stimulation (DBS) guided by intracranial recordings is funded by the NIH BRAIN Initiative (UH3 NS103549) and approved by the Institution Review Board at Baylor College of Medicine (IRB number H-43036). These individuals did not carry significant psychiatric comorbidities based on the trial's exclusion of schizophrenia, bipolar disorder, personality disorders, and neuro-developmental disorders, as these conditions may impact the study results. Additional details regarding the exclusion criteria can be found: https://clinicaltrials.gov/study/NCT03437928#participation-criteria. Each patient was implanted with permanent deep brain stimulation leads for stimulation delivery as well as with temporary sEEG electrodes for neural recordings. In our study, two patients used antidepressant medication, while the remaining two patients did not receive any medication. The trial protocol requires patients to maintain a stable dose of medication for at least one month before surgery, and no alterations are made to their medication during the in-patient monitoring period.

In this study, we focus on the sEEG recordings from reward-relevant regions. Following the surgical implantation of electrodes, patients underwent around one week of inpatient monitoring. Throughout this time, we conducted probabilistic reward task while simultaneously recording dense neural activity.

### Probabilistic reward task

Probabilistic reward task (PRT) is a task for the investigation of reward processing. Each run of the PRT consists of 300 trials, divided into 3 blocks of 100 trials. All three blocks were performed on the same day, and a short break was allowed between the blocks. Each trial started with the presentation of a fixation cross in the center of the screen for a random duration ranging from 700 to 900 ms. A mouthless cartoon face was then presented for 500 ms followed by the presentation of this face with either a short mouth or a long mouth for 100 ms. Participants were then asked to identify which type of mouth was presented. Either reward or neutral feedback was presented for 1000 ms after a random delay interval ranging from 500 to 1000 ms. For each run, we set one type of stimulus (either the short or long mouth) as the rich stimulus. The probability of reward for rich stimulus is three times more frequent than for lean stimulus (the other kind of mouth). If a participant didn't correctly identify the stimulus in a trial where reward feedback was scheduled, the feedback was postponed until the next accurate identification of the same stimulus type. Therefore, the consistency of reward feedback frequency was maintained across all participants for each stimulus type. This approach is the same as that used in the earlier study with the same task[5]. The likelihood of receiving a reward was set at either 60% or 20%, depending on whether the choice was categorized as the rich stimulus or the lean stimulus. Reaction time was defined as the time between stimulus presentation and participant response. As a previous study using PRT[5], trials with reaction time that were too short (<150 ms), too long (>2500 ms), or were considered an outlier response (exceeding mean ± 3 SD), were excluded from further analysis. The task was implemented in MATLAB (The Mathworks, Inc., Natick,

MA) using Psychtoolbox-3. For behavioral analyses on the PRT task, the main variable of interest is response bias, which measures the systematic preference for the response associated with more frequently rewarded (rich) stimulus. Response bias can be calculated based on the number of correct or incorrect trials to the rich and lean stimuli.

$$\log b = \frac{1}{2}\log \frac{rich_{correct} * lean_{incorrect}}{rich_{incorrect} * lean_{correct}} \qquad (1)$$

Previous studies suggest that response bias is inversely related to current anhedonic symptoms in unselected adults. It is also blunted in depression patients and is improved by pharmacological treatments[53,54].

### Electrode localization

Patients underwent brain magnetic resonance imaging (MRI) before surgery and computed tomography (CT) after implantation. To identify the precise anatomical position of implanted electrodes, we co-registered the pre-operative T1-weighted MRI scans with the postoperative CT scans. Automatic cortical reconstruction was performed on the preoperative T1-weighted MRI using Freesurfer tools[93]. Functional Magnetic Resonance Imaging for the Brain Software Library's Linear Image Registration Tool (FLIRT) was employed to align the postoperative CT data with the preoperative T1-weighted MRI[94]. Electrode positions were manually marked using the co-registered CT data in BioImage Suite v3.5b1 and plotted into the native MRI space[95]. An expert reviewer then examined the images to identify brain regions and determine whether the contact was in gray or white matter. Contacts determined to be in white matter were excluded from further analysis. Detailed procedures have been described elsewhere[62,65].

### Data acquisition and preprocessing

Neural signals were recorded during the probabilistic reward task. Signals were recorded with sEEG electrodes at 2000 Hz using a Cerebus data acquisition system (BlackRock Microsystems, UT, USA) with a bandpass of 0.3–500 Hz (4th order Butterworth filter). We visually inspected raw signals for the presence of recording artifacts and interictal epileptic spikes. Channels with excessive noise were excluded to prevent noise from spreading to other channels through re-reference. Signals were notch filtered (60 Hz and its harmonics) to reduce line-noise artifacts and re-referenced through bipolar reference to reduce the effects of volume conduction[96]. We then down-sampled referenced signals to 1000 Hz.

### Spectral power analysis

We performed Hilbert transform to estimate spectral power in six different frequency bands: 1–4 Hz (delta), 4–8 Hz (theta), 8–12 Hz (alpha), 12–30 Hz (beta), 35–50 Hz (gamma) and 70–150 Hz (high-gamma). Spectral power values for each trial were calculated by averaging the squared magnitude of the Hilbert transform decomposition. For the delay period, spectral power was averaged across a time window beginning at behavioral response and ending at 500 ms after choice. For the feedback period, the mean spectral power was calculated across a time window beginning at feedback onset and ending at feedback offset, which was set at 1000 ms after feedback onset. The spectral power values were then normalized as a percent change relative to the baseline, which was the fixation cross period at the beginning of each trial. To ensure data quality, any value that deviated by more than three standard deviations from the mean value was considered an outlier and excluded from further analysis. Lastly, the values were z-scored within each contact for visualization.

### Data visualization in the template brain

Contacts were projected onto the MNI space and visualized using the open-source software RAVE (R Analysis and Visualization of iEEG)[97].

To analyze the data during the delay period, a two-sample $t$ test was employed to generate a $t$ value for each contact. These $t$ values, which compared the average spectral power after the rich and lean stimuli, were then plotted onto the template brain. Similarly, for the analysis of the data during the feedback period, a two-sample $t$ test was used to compare reward feedback with neutral feedback, and the resulting $t$ values were plotted at each contact.

## Linear mixed effect model

Statistical analyses on data from all channels mainly focused on the effects of different conditions on the spectral power change. We used a linear mixed effect model to quantify the effect of stimulus type or feedback type on the power change. We modeled stimulus type or feedback type as a fixed effect, and channels and participants as nested random effects. The formula for neural activity during the delay period is Power ~ stimulus + (1|Subject/chan) while The formula for neural activity during the feedback period is Power ~ feedback + (1|Subject/chan). This model was used for the control group and the depression group separately. To further test whether the power change was modulated by the group type, we included both the group type and the feedback type as fixed effects in the linear mixed effect model and calculated the corresponding $p$ value. The formula used in this analysis is Power ~ feedback+grouptype+feedback*grouptype + (1|Subject/chan). The $p$ value associated with each parameter is derived from a two-sided test.

## Reporting summary

Further information on research design is available in the Nature Portfolio Reporting Summary linked to this article.

## Data availability

Processed data is provided at the following address: https://osf.io/t3usq/. The raw intracranial EEG data are available upon request for reasons of patient confidentally. Source data are provided with this paper.

## Code availability

Code used in this study is available under: https://osf.io/t3usq/.

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

## Acknowledgements
We thank our study participants for their commitment and trust. This work was supported by the National Institutes of Health (Grant No. UH3 NS103549 [to S.A.S., K.R.B., W.K.G., and N.P.], Grant No. K01 MH116364 [to K.R.B.], Grant No. R21 NS104953 [to K.R.B.], Grant No. UH3 NS100549 [to W.K.G. and S.A.S.], and Grant No. R01 MH114854 [to W.K.G.]) and McNair Foundation (to S.A.S.).

## Author contributions
J.X. and S.A.S. conceptualized and designed the study. J.X. performed data collection, analyzed the data, and drafted the manuscript with support from J.A.A., J.M., A.B.A., R.K.M., V.P., R.N., N.R.P., E.B., A.J.W., D.O., R.G., A.A., and B.S. S.A.S., N.P., W.K.G., S.J.M., and K.R.B. oversaw the organization of the clinical trial, subject recruitment, and regulatory activities. B.H., X.P., and S.A.S. supervised and guided the data analysis. J.X., B.H., and S.A.S. wrote and edited the manuscript, and all authors contributed to review and revision of the manuscript.

## Competing interests
S.A.S. has consulting agreements with Boston Scientific, NeuroPace, Abbott, and Zimmer Biomet. W.K.G. receives royalties from Nview, LLC and OCDscales, LLC. S.J.M. has served as a consultant to the following companies: Almatica Pharma, Biohaven, BioXcel Therapeutics, Boehringer-Ingelheim, Brii Biosciences, Clexio Biosciences, COMPASS Pathways, Delix Therapeutics, Douglas Pharmaceuticals, Eleusis, Engrail Therapeutics, Freedom Biosciences, Janssen, Liva Nova, Levo Therapeutics, Merck, Neumora, Neurocrine, Perception Neurosciences, Praxis Precision Medicines, Relmada Therapeutics, Sage Therapeutics, Seelos Therapeutics, Signant Health, Sunovion, Xenon Pharmaceuticals, and XW Pharma. S.J.M. has received research support from Boehringer-Ingelheim, Engrail Therapeutics, Merck, Neurocrine, and Sage Therapeutics. N.P. is a consultant for Abbott Laboratories and Sensoria Therapeutics. The remaining authors declare no competing interests.
