## [Peer Review File · Nature Communications]

Beta activity in human anterior cingulate cortex mediates reward biasesREVIEWER COMMENTS

Reviewer #1 (Remarks to the Author):

Sheth and colleagues in Beta activity in human anterior cingulate cortex mediates reward biases present a well constructed and interesting paper with novel findings.

One relevant point for this group is that diminution in reward biasing is a hallmark of clinical depression.

Their group hypothesized that reward biasing was mediated by the anterior cingulate cortex (ACC), a cortical hub region associated with the integration of reward and executive control and with the etiology of depression.

A key finding was that beta (12-30 Hz) oscillations in the ACC predicted both associated reward and the size of the choice bias, and also tracked reward receipt, thereby predicting bias on future trials.

ACC

Their notion that beta oscillations may orchestrate the learning of reward information to guide adaptive choice, and, more broadly, suggest a potential biomarker for anhedonia and point to future development of interventions to enhance reward impact for therapeutic benefit is an over-reach and not supported by the limited data.

The paper has only a small number of patients and epilepsy patients may not be a normative comparator.

The paper focuses on the perceptual bias task only and this task has been used in anhedonia only limiting the more general interpretation.

The idea of this being a temporal dynamic signal is exciting.

Tone this statement/claim down: This finding positions ACC beta activity as the pivotal element connecting rewards to broader cognitive domains, potentially offering novel avenues for enhancing cognitive and emotional well-being.

There were only 4 patients in the depression group making the findings potentially tenuous.

Should the title more appropriately be: Beta activity in human anterior cingulate cortex during an anhedonia task and reward biases.

Reviewer #2 (Remarks to the Author):

The authors provide a provocative analysis of intracranial activity related to biasing effects using patients with either epilepsy or treatment resistant depression undergoing SEEG recordings in preparation for further treatment. Specifically, they find that beta activity is enhanced with biasing in the ACC. However, this response is blunted in patients with depression vs. those with epilepsy (and no mood disorder). This is an important finding, if it holds up, and tells us a potential electrophysiological marker corresponding to differences in reward learning, which is diminished in depression.

I am worried about the statistical validity of some of the findings, though. First, the authors used a linear mixed effects model to find significant power changes as a function of stimulus type. The exact nature of the model is not totally clear. The authors should provide a more formal definition of the model, for example using Wilkinson notation, so that readers and reviewers can better understand exactly what statistical tests they were running.

Next, the p-values they provide are modest for many findings (like 0.039) and it is unclear if there was any control for multiple comparisons. If they ran an individual LME for each possible frequency band, then picked the significant frequency band from that population of multiple tests, that increases the chance of a false positive significantly. For example, on line 136, they show a p-value of 0.039 for the beta power LME for ACC. If this was done for each power band they analyze (of which there are 6), and the authors do not correct for multiple comparisons, and use a p-value threshold of 0.05, the chance of a false positive is ~26%. A simple Bonferonni correction would argue for a p-value threshold of $< \sim 0.0083$ (which is far lower than the 0.039 the authors report). This might even be worse if the authors are using LMEs for each anatomical region, but that is hard to discern without a better formal definition of their LME (see previous point).

The same problem arises again in Fig. 3D, where 6 regressions are run without controlling for multiple comparisons. The reported single significant p-value of 0.013 is > the Bonferroni corrected p-value of 0.0083. Again, the risk of a false positive is ~26% with this approach, without correcting for multiple comparisons.

Another concern is how robust these findings are for the beta frequency specifically. The authors pick ranges for different frequency bands (like 12-30 Hz for beta) that are somewhat arbitrary (arbitrary since there are many other ranges used for beta in the literature). If the authors' finding is a robust finding, the precise frequency range should have a tolerance for alternative definitions of beta, like 13-29.5 Hz (Greer et al., 2021), 22-28 Hz (Palacios-Garcia et al., 2021), 14-30 Hz (Akbulut et al., 2019), or 13-30 Hz (Spitzer and Haegens, 2017), yielding qualitatively similar results regardless of the precise range used. In engineering or mathematical modeling, this is called a sensitivity analysis. In other words, the results shouldn't be so fragile that slight changes to arbitrary parameters (in this case the frequency band of beta) cause the results to become non-significant. I bet the authors results will be fine if they do a sensitivity analysis, but it would be further confirmation that this is a real phenomenon, and not a statistical artifact. They should do it.

Minor points:

The Methods section states that none of the epilepsy patients had a mood disorder. How was this objectively assessed?

For p-value reporting, why not put the actual p-values? For example, line 108, line 110... They do this successfully in other parts of the paper, like line 136 and line 145.

Reviewer #3 (Remarks to the Author):

Summary

This is well well-written and valuable manuscript investigating the neurophysiological correlates of reward choices in a unique data set of patients with major depressive disorder and a control group of epilepsy patients. The authors measured intracranial EEG (ACC, mOFC, amygdala) associated with a stochastically rewarded perceptual decision task in both patient groups. The behavioral data analysis demonstrated that responses to highly rewarding stimuli were more accurate compared to responses to less rewarding stimuli (this suggests a response bias) in the epilepsy patients. This response bias was blunted in the depression group. The epilepsy patients also showed increased reward anticipation in the delay period, as evidenced by increased beta activity in the ACC with rich compared to lean stimuli. This was additionally correlated with the enhanced response bias.

Likewise, the receipt of rewarding feedback was linked with an increase in beta. The depression group did not demonstrate the reward-related increase in beta activity in the ACC, neither during the delay period nor with the reward feedback. Both findings suggest a blunted reward response in the depression group.

My questions and comments are mainly regarding the (choices for specific) analyses of the data.

Data analysis and interpretation:

1. Did the authors analyze feedback-related activity in other frequency bands than beta and if so, what were the results for each group? If these were all null findings, please provide this in a footnote or supplementary section. Similarly, what were the findings for frequencies other than beta during the delay period in the depression group (were these null findings similar to the epilepsy group)?

If other frequencies were not analyzed, please provide an explanation for the focus on the beta band in the introduction, especially since the reward learning literature has frequently demonstrated the role of theta band activity in the ACC during reward-based learning, i.e. prediction errors and reward positivity (see for example Cavanagh & Frank, 2014; Lin et al. 2018)

2. Regarding the prediction error: Was there a relation between delay period beta activity (anticipation for a reward with rich vs lean stimuli) and the actual feedback related activity (either in beta or another frequency)? That is, did the size of the delay period activity impact the potential feedback-related prediction error when expectations were violated and was this different between the groups? See for example Unemoto & Holroyd, 2017) for the relation between reward anticipation and individual differences in depression.

3. Could the authors show the behavioral data in the depression group, i.e. what was the accuracy on the task in the depression cohort (separate for rich and lean stimuli) and what was their response bias across blocks?

4. Related to the previous question; if there was a difference in accuracy between the groups, how did this impact the frequency of reward feedback between the group? A higher reward frequency could impact the anticipation of a positive outcome over time and the associated beta activity in the delay period. Did the authors check the relative amount of reward received in relation to the size of the beta response during the delay period for each group?

Fig 5c: the beta power in the depression group seems to be almost null for both stimulus types, which makes it difficult to detect a difference between conditions, or to compare this data with the epilepsy group. Is the absence of beta in general related to depression (regardless of condition/task)? Please explain or elaborate on this in the discussion.

Discussion:

Overall, the discussion would benefit from explaining the current findings in the context of the previous literature on the neural response to rewarding stimuli with depression, i.e. the reward-related beta increase in ACC versus theta in previous studies. Also, previous behavioral findings may be less consistent than what is discussed here (line 292, 293); there are several studies that did not find a behavioral difference in reward learning between healthy controls and depressive cohort. Please include a discussion of those findings (see for example Davide et al. 2019 or Rothkirch et al. 2017)

1. Lin, H., Saunders, B., Hutcherson, C. A., & Inzlicht, M. (2018). Midfrontal theta and pupil dilation parametrically track sub-jective conflict (but also surprise) during intertemporal choice. *NeuroImage*, 172, 838– 852.
2. Cavanagh, J. F., & Frank, M. J. (2014). Frontal theta as a mechanism for cognitive control. *Trends in Cognitive Sciences*, 18(8), 414– 421
3. Umemoto, A., & Holroyd, C. B. (2017). Neural mechanisms of re-ward processing associated with depression- related personality traits. *Clinical Neurophysiology*, 128(7), 1184– 1196.
4. Abnormal approach-related motivation but spared reinforcement learning in MDD: Evidence from fronto-midline Theta oscillations and frontal Alpha asymmetry. Davide Gheza , Jasmina Bakic, Chris Baeken , Rudi De Raedt , Gilles Pourtois *Cogn Affect Behav Neurosci* . 2019 Jun;19(3):759-777. doi: 10.3758/s13415-019-00693-4.
5. Neural mechanisms of reinforcement learning in unmedicated patients with major depressive disorder Marcus Rothkirch Jonas Tonn , Stephan Köhler, Philipp Sterzer. *Brain*. 2017 Apr 1;140(4):1147-1157. doi: 10.1093/brain/awx025.

Reviewer #4 (Remarks to the Author):

The study by Xiao et al report an analysis on the involvement of beta oscillations in the anterior cingulate cortex (ACC) in modulating reward biases within the context of a probabilistic reward task. The findings revealed an increase in ACC beta power subsequent to the selection of a more frequently rewarded stimulus, and this was found to be positively associated with individual variations in behavioral biasing. These outcomes are taken to suggest that ACC beta oscillations may serve as indicators of reward learning processes. The paper is not only well-structured and easily comprehensible but also establishes a compelling rationale linking ACC functionality to the realms of reward and depression. The results presented lend strong support to the stated

conclusions, rendering this paper a valuable addition to the existing body of knowledge on the neural underpinnings of reward processing and anhedonia. Nevertheless, I do have some specific comments on the manuscript, which I hope will aid the authors to improve the quality of their reporting – as detailed below:

Major comments

1. The authors may want to provide a rationale for their choice of investigating the beta band and substantiate its relevance to their research question by referencing prior literature.
2. In the case of an exploratory analysis, it is essential to transparently discuss null results, including t-values, 95% confidence intervals, and p-values for other frequency bands. It is recommended to report the R-squared value of the mixed-effect model to ensure the statistical validity of the reported results. Please also include information about the residual distribution of the mixed-effect model to confirm the model's appropriateness. For each statistical test, please provide details on the degrees of freedom and the lower and upper bounds of the 95% confidence intervals.
3. Please specify the number of permutations used for the cluster-corrected analysis.
4. Specify the number of patients with electrode coverage in each of the four recording sites for both the epileptic and depression cohorts. Given the small size of the depressed cohort (N=4), consider discussing the challenges of attributing differences solely to depression and explore individual-level effects.
5. Disclose whether the depression cohort was using any antidepressant medication and how the results were or were not different across the two groups. If medication was used, conduct an analysis to investigate the relationship between ACC beta power and duration and medication dosage. Please also ensure that the attenuated effect in the depression cohort is specific to the beta band in the ACC.
6. Validate that the depression cohort is free of comorbidities with other disorders, establishing it as a pure major depressive disorder cohort.
7. While the authors suggest ACC beta as a biomarker, it's important to discuss the feasibility of translating these findings to non-invasive EEG/MEG methods.
8. Clarify how you ruled out the possibility that the observed increase in beta values during both anticipation and receipt of rewards was not influenced by general decision-related processing or cognitive effort.
9. Lastly, pls consider commenting on the precise anatomical location of the effects (within ACC) and comment on prior electrical stimulation studies linking dACC and motivation/drive (e.g., Parvizi, Neuron 2013 and Fox et al Nature Human Behavior 2020)

We would like to thank the Editor for the consideration of our manuscript for publication, and the Reviewers for their insightful and constructive comments. Our response is below with Reviewer comments in black, our responses in blue, text that was in the original submission in *blue italics*, and text revisions in red. We believe the impact has improved considerably with these revisions and hope the revised manuscript is now suitable for publication.

Reviewer #1

Sheth and colleagues in Beta activity in human anterior cingulate cortex mediates reward biases present a well constructed and interesting paper with novel findings. One relevant point for this group is that diminution in reward biasing is a hallmark of clinical depression. Their group hypothesized that reward biasing was mediated by the anterior cingulate cortex (ACC), a cortical hub region associated with the integration of reward and executive control and with the etiology of depression. A key finding was that beta (12-30 Hz) oscillations in the ACC predicted both associated reward and the size of the choice bias, and also tracked reward receipt, thereby predicting bias on future trials.

We appreciate these positive comments.

Their notion that beta oscillations may orchestrate the learning of reward information to guide adaptive choice, and, more broadly, suggest a potential biomarker for anhedonia and point to future development of interventions to enhance reward impact for therapeutic benefit is an over-reach and not supported by the limited data.

The paper has only a small number of patients and epilepsy patients may not be a normative comparator.

We thank the reviewer for this thoughtful comment. We had a total of 15 epilepsy patients and 4 treatment-resistant depression patients in our study. The overall number of human subjects included is large compared to most studies using intracranial neural recordings, and the inclusion of 4 TRD patients' worth of data from these diverse brain regions is truly unique.

The papers below include eight to thirteen epilepsy patients without other cohorts of patients that have intracranial recording:

- Huang, Y. *et al.* The insulo-opercular cortex encodes food-specific content under controlled and naturalistic conditions. *Nat. Commun.* **12**, 3609 (2021).
- Collomb-Clerc, A. *et al.* Human thalamic low-frequency oscillations correlate with expected value and outcomes during reinforcement learning. *Nat. Commun.* **14**, 6534 (2023).
- Johnson, E. L. *et al.* A rapid theta network mechanism for flexible information encoding. *Nat. Commun.* **14**, 2872 (2023).
- ter Wal, M. *et al.* Theta rhythmicity governs human behavior and hippocampal signals during memory-dependent tasks. *Nat. Commun.* **12**, 7048 (2021).
- Schreiner, T., Kaufmann, E., Noachtar, S., Mehrkens, J.-H. & Staudigl, T. The human thalamus orchestrates neocortical oscillations during NREM sleep. *Nat. Commun.* **13**, 5231 (2022).

We acknowledge that the number of TRD patients (N=4) is low. However, the difference between the depression group and the epilepsy group is statistically significant (Figure 5). The low number of patients is due to the nature of the clinical trial of deep brain stimulation for severe treatment-resistant depression that uses an inpatient intracranial platform for therapy development. It is a rare opportunity to conduct intracranial recording in patients with neuropsychiatric disorders. These studies usually have very small cohort sizes due to their invasive nature. This number of patients is sufficient to obtain statistical significance when the effects are highly consistent. Indeed, this fact is widely recognized in the field: below is a list of papers that include the same number of patients or fewer with a specific type of neuropsychiatric disorder:

One patient with treatment-resistant depression:

- Scangos, K. W. *et al.* Closed-loop neuromodulation in an individual with treatment-resistant depression. *Nat. Med.* **27**, 1696–1700 (2021).
- Scangos, K. W., Makhoul, G. S., Sugrue, L. P., Chang, E. F. & Krystal, A. D. State-dependent responses to intracranial brain stimulation in a patient with depression. *Nat. Med.* **27**, 229–231 (2021).
- Scangos, K. *et al.* Corticocortical Evoked Potentials and Patient Response Reveal Networks Underlying Depression. *Biol. Psychiatry* **87**, S157 (2020).

One patient with OCD:

- Nho, Y.-H. *et al.* Responsive deep brain stimulation guided by ventral striatal electrophysiology of obsession durably ameliorates compulsion. *Neuron* **112**, 73-83.e4 (2024).

Two patients with PTSD:

- Gill, J. L. *et al.* A pilot study of closed-loop neuromodulation for treatment-resistant post-traumatic stress disorder. *Nat. Commun.* **14**, 2997 (2023).

Five patients with OCD:

- Provenza, N. R. *et al.* Long-term ecological assessment of intracranial electrophysiology synchronized to behavioral markers in obsessive-compulsive disorder. *Nat. Med.* **27**, 2154–2164 (2021).

Four patients with refractory neuropathic pain:

- Shirvalkar, P. *et al.* First-in-human prediction of chronic pain state using intracranial neural biomarkers. *Nat. Neurosci.* **26**, 1090–1099 (2023).

Our study includes both epilepsy patients and TRD patients, making it stand out in terms of volume and diversity of diagnosis even across these published studies.

The paper focuses on the perceptual bias task only and this task has been used in anhedonia only limiting the more general interpretation.

It is true that this task is used primarily for anhedonia. Indeed, we see that as a major virtue of the study. We believe that our original manuscript did not sufficiently clarify the importance of

anhedonia and its promise as an avenue for improving diagnosis and treatment of depression (see, for example, Cooper et al., 2018; Vrieze et al., 2013; Wang et al., 2021). As such there is a lot of interest in understanding it. The perceptual bias task is the most well-validated and well-studied task for anhedonia (Insel et al., 2010; Kangas et al., 2020; Pizzagalli et al., 2008). Moreover, it has not only been used for anhedonia, but also as a more general task to assess reward learning and reward-related activation within the neural circuitry in both rodents and humans (Der-Avakian et al., 2013; Insel et al., 2010; Iturra-Mena et al., 2023; Sailer et al., 2023; Wilkinson et al., 2021). We have added the following new text to the revised Discussion making these points clear:

Anhedonia, with its profound impact on an individual's quality of life, can create challenges in various aspects including relationships, work, and daily functioning. Traditional antidepressants often fail to adequately address this symptom (Vrieze et al., 2013; Wang et al., 2021). Thus, gaining a deeper understanding of anhedonia is important for improving diagnosis and treatment of depression. *The perceptual bias task is the most commonly used task for the assessment of anhedonia, a major element of clinical depression. There are two reasons for this: first, the task has direct face validity to the association of reward and action; second, there is ample empirical evidence linking changes in behavior in this task with anhedonia.* It also has been used to assess reward learning and feedback responsiveness in both rodents and humans (Der-Avakian et al., 2013; Iturra-Mena et al., 2023; Sailer et al., 2023; Wilkinson et al., 2021).

The idea of this being a temporal dynamic signal is exciting.
We appreciate this positive comment.

Tone this statement/claim down: This finding positions ACC beta activity as the pivotal element connecting rewards to broader cognitive domains, potentially offering novel avenues for enhancing cognitive and emotional well-being.

We revised this statement, which is the last sentence of our manuscript, to the following sentences:

This finding suggests that ACC beta activity could play a key role in linking rewards to various cognitive areas. Future studies might explore how altering this activity could impact cognitive and emotional well-being.

There were only 4 patients in the depression group making the findings potentially tenuous.

As mentioned earlier, the limited patient enrollment is a result of the unique clinical trial that utilizes an inpatient intracranial platform for the development of depression therapy. The inherent invasiveness of these studies generally leads to smaller cohort sizes. We have added the text below to acknowledge the limitation of only four patients in the depression group and the need for studies with a larger population in the future. On the other hand, the statistical significance of the results warrants their inclusion.

Such a biomarker has many potential benefits, including the ability to improve diagnosis and symptom monitoring. Moreover, they present an appealing target for neuromodulatory trials, which could focus on altering ACC beta and thereby reducing anhedonia. **The limited**

enrollment of only four depression patients in our study is due to the unique clinical trial using an inpatient intracranial platform for therapy development. Future studies with larger sample sizes will be necessary to extend the generalizability of our findings to the broader population of individuals with depression.

Should the title more appropriately be: Beta activity in human anterior cingulate cortex during an anhedonia task and reward biases.

We appreciate the reviewer's suggestion. However, we feel the more precise terminology ("reward bias task") is better due to its precision. Moreover, as we note above, this is not strictly speaking an anhedonia task - first, it indexes anhedonia although it doesn't measure anhedonia per se; second, it has other uses besides indexing anhedonia.

We believe that labeling it as an "anhedonia task" may not precisely capture its conceptual scope, as it encompasses broader aspects. While we considered a more detailed title like "Beta activity in the human anterior cingulate cortex mediating reward biases in a task with demonstrated relevance for anhedonia in depression", we think that is overly long. Our study primarily investigates basic neural mechanisms in psychiatrically healthy individuals compared to a specific population of treatment-resistant depression patients. Therefore, the relevance to anhedonia is indirectly applicable to the majority of our cohort. Consequently, we would prefer to keep the original title, 'Beta activity in human anterior cingulate cortex mediates reward biases'.

Reviewer #2

The authors provide a provocative analysis of intracranial activity related to biasing effects using patients with either epilepsy or treatment resistant depression undergoing SEEG recordings in preparation for further treatment. Specifically, they find that beta activity is enhanced with biasing in the ACC. However, this response is blunted in patients with depression vs. those with epilepsy (and no mood disorder). This is an important finding, if it holds up, and tells us a potential electrophysiological marker corresponding to differences in reward learning, which is diminished in depression.

I am worried about the statistical validity of some of the findings, though. First, the authors used a linear mixed effects model to find significant power changes as a function of stimulus type. The exact nature of the model is not totally clear. The authors should provide a more formal definition of the model, for example using Wilkinson notation, so that readers and reviewers can better understand exactly what statistical tests they were running.

We thank the reviewer for this thoughtful comment and suggestion. For the neural activity during the delay period, we used stimulus type (rich vs. lean) as the predictor variable. For the neural activity during the feedback period, we used feedback type (reward vs. no reward) as the predictor variable. To compare neural activity between depression patients and epilepsy patients, we added the group type and the interaction term as the predictor variables. We have added the

Wilkinson notation to all linear mixed effects models used in this study in the revised Methods to enhance the clarity and transparency of statistical analyses:

Linear mixed effect model

*Statistical analyses on data from all channels mainly focused on the effects of different conditions on the spectral power change. We used a linear mixed effect model to quantify the effect of stimulus type or feedback type on the power change. We modeled stimulus type or feedback type as a fixed effect, and channels and participants as nested random effects. The formula for neural activity during the delay period is $\text{Power} \sim \text{stimulus} + (1|\text{Subject}/\text{chan})$ while the formula for neural activity during the feedback period is $\text{Power} \sim \text{feedback} + (1|\text{Subject}/\text{chan})$. This model was used for the control group and the depression group separately. To further test whether the power change was modulated by the group type, we included both the group type and the feedback type as fixed effects in the linear mixed effect model and calculated the corresponding p-value. The formula used in this analysis is $\text{Power} \sim \text{feedback} + \text{grouptype} + \text{feedback} * \text{grouptype} + (1|\text{Subject}/\text{chan})$.*

Next, the p-values they provide are modest for many findings (like 0.039) and it is unclear if there was any control for multiple comparisons. If they ran an individual LME for each possible frequency band, then picked the significant frequency band from that population of multiple tests, that increases the chance of a false positive significantly. For example, on line 136, they show a p-value of 0.039 for the beta power LME for ACC. If this was done for each power band they analyze (of which there are 6), and the authors do not correct for multiple comparisons, and use a p-value threshold of 0.05, the chance of a false positive is ~26%. A simple Bonferroni correction would argue for a p-value threshold of $< \sim 0.0083$ (which is far lower than the 0.039 the authors report). This might even be worse if the authors are using LMEs for each anatomical region, but that is hard to discern without a better formal definition of their LME (see previous point).

The same problem arises again in Fig. 3D, where 6 regressions are run without controlling for multiple comparisons. The reported single significant p-value of 0.013 is $>$ the Bonferroni corrected p-value of 0.0083. Again, the risk of a false positive is ~26% with this approach, without correcting for multiple comparisons.

Thank you for the important comment. In analyses of local field potentials, there are six bands of activity in which meaningful responses could potentially be found. If the *a priori* hypothesis is that activity in any of these bands would be considered meaningful - that is, if such activity would validate the hypothesis, then Bonferroni correction is mandatory, and as a result the alpha criterion for statistical significance must be lower than 0.05 to achieve a p-value equivalent to $p=0.05$. On the other hand, if the prior hypothesis is that only one of those bands would validate the hypothesis, then the Bonferroni correction would be incorrect, and deploying it would be overly conservative and lead to Type II errors. In the case of our paper, there was a natural and strong prior hypothesis - namely, that activity would be observed in the beta band. This prior hypothesis comes from the beta increase observed after informing participants about the monetary gains (Cohen et al., 2007; HajiHosseini et al., 2012; HajiHosseini & Holroyd, 2015; Marco-Pallares et al., 2008; Marco-Pallarés et al., 2015). It also comes from the findings of a

recent paper performing chronic recordings in a closely connected region, the subcallosal cingulate, in individuals with TRD (Alagapan et al., 2023).

As a result, the anticipated finding was in the beta band, and Bonferroni correction would be invalid. The reason we present the other bands is for thoroughness. In this case, there is no specific prior in favor or against those five bands, and as a result, we agree with the reviewer that Bonferroni correction is required - a criterion using an alpha of 0.05 would have a substantially greater than 5% chance of turning up a false positive. Thus, we have made two changes to the manuscript to reflect this difference. First, we have recast the Introduction and Results to emphasize that our core experimental hypothesis was in favor of the beta band, and we have given stronger justification for that claim. Second, we have now emphasized that the remaining five bands are tested for reasons of completeness and have included a Bonferroni correction for them. We thank the reviewer for bringing up the important concern.

Several human electrophysiological studies have identified a mediofrontal oscillatory component associated with positive feedback in both gambling task and reversal learning task, tasks that have key features in common with our bias task. The increase observed in these tasks is in the beta range and occurs 200 to 400 ms after the feedback informing the participant about the monetary gains (Cohen et al., 2007; HajiHosseini et al., 2012; HajiHosseini & Holroyd, 2015; Marco-Pallares et al., 2008; Marco-Pallarés et al., 2015). It has been proposed that this beta activity is generated in the prefrontal cortex; the most commonly inferred source site is the dACC. It is further assumed that the ACC then transmits a fast motivational value signal, still in the beta band, from the frontal cortex to downstream reward-related regions (Marco-Pallarés et al., 2015). Moreover, beta activity in the cingulate cortex is of particular importance in depression (Clark et al., 2016; Huebl et al., 2016; Merkl et al., 2016). For example, a recent study showed that beta activity best tracks depressive states, seen as a decrease in beta band power during the first month of chronic stimulation, followed by an eventual rise (Alagapan et al., 2023). This result suggests that sustained, antidepressant responses might involve increased beta band power after prolonged stimulation.

We first analyzed the delay period (Fig. 2a), which extends 500 msec after choice. During this period, sensory information and reward history can be integrated to generate value representations for stimuli, enabling the brain to anticipate the potential rewards and potentially promote learning. We computed the average spectral power at six major frequencies (delta, theta, alpha, beta, gamma, and high-gamma) during the delay period (using the same definitions as in Xiao et al., 2023). Considering the observation of beta activity during reward processing and the importance of beta activity in depression, our core experimental hypothesis was in favor of the beta band. Results from the other five bands with Bonferroni correction were reported for reasons of completeness.

On the other hand, these effects were not brain-wide. Specifically, the beta modulation in medial orbitofrontal cortex and amygdala was not significant ($t(\text{stimulus}) = -1.7, p = 0.09, \text{coef} = -5.1, 95\% \text{ confidence interval (95\% CI)} = [-12.8 - 2.7]$, for mOFC beta, $t(\text{stimulus}) = -1.3, p = 0.20, \text{coef} = -2.1, 95\% \text{ confidence interval (95\% CI)} = [-4.4 - 0.3]$, for amygdala beta, before multiple comparison corrections). Note that the failure to achieve significance is not itself dispositive; indeed, we report other evidence linking mOFC and amygdala in associations, suggesting that ACC may be part of a larger network focused on learning (see below).

These results clearly implicate ACC beta activity in processing related to biasing. We found some evidence that OFC is likewise involved in reward learning. Response bias was positively correlated with the difference in beta power response towards rich or lean stimulus in the orbitofrontal cortex ($p = 0.065$ for mOFC beta, $p = 0.037$ for lOFC beta before multiple comparison corrections).

We found evidence that ACC is not alone in this process; indeed, two other regions from which we recorded, mOFC and amygdala, also showed a systematic relationship between post-reward beta activity and reward (mOFC: $t(\text{feedback}) = -2.0$, $p = 0.04$, coef = -3.4, 95% confidence interval (95% CI) = [-6.8 – -0.09], amygdala: $t(\text{feedback}) = -5.0$, $p < 10^{-4}$, coef = -9.8, 95% confidence interval (95% CI) = [-13.6 – -6.0], before multiple comparison corrections). It is interesting that amygdala showed a significant reversed relationship even after multiple comparison corrections; only ACC showed the same, positive, relationship during both the post-choice and delay periods.

Our study reveals that beta oscillations in the ACC, which represents reward outcome, are also elicited by rich stimuli and are correlated with behavioral preference. These findings provide further support for the importance of the ACC in reward learning. Indeed, these results are consistent with previous theories linking the ACC with cognition related to reward in general and to reward-mediated learning specifically. In experiments involving both monetary gains and losses, research has demonstrated that an increase in theta power is associated with losses, while an increase in beta power is associated with gains (Cohen et al., 2007; HajiHosseini & Holroyd, 2015; Marco-Pallares et al., 2008). Aligned with prior findings, our study showed a rise in beta activity within the ACC following positive feedback. Future studies could investigate ACC activity during a similar task that incorporates negative feedback for a more comprehensive understanding.

Another concern is how robust these findings are for the beta frequency specifically. The authors pick ranges for different frequency bands (like 12-30 Hz for beta) that are somewhat arbitrary (arbitrary since there are many other ranges used for beta in the literature). If the authors' finding is a robust finding, the precise frequency range should have a tolerance for alternative definitions of beta, like 13-29.5 Hz (Greer et al., 2021), 22-28 Hz (Palacios-Garcia et al., 2021), 14-30 Hz (Akbulut et al., 2019), or 13-30 Hz (Spitzer and Haegens, 2017), yielding qualitatively similar results regardless of the precise range used. In engineering or mathematical modeling, this is called a sensitivity analysis. In other words, the results shouldn't be so fragile that slight changes to arbitrary parameters (in this case the frequency band of beta) cause the results to become non-significant. I bet the authors results will be fine if they do a sensitivity analysis, but it would be further confirmation that this is a real phenomenon, and not a statistical artifact. They should do it.

It is true that there are different ranges used for beta in the literature, although, as the reviewer points out, many are fairly similar. However, the range that we use here (12-30Hz) is widely used in the field and considered by many to be the closest approximation to a "standard" definition. Moreover, this range was used in several of the studies that motivated our own, including Babapoor-Farrokhran et al., 2017; Das et al., 2022; Ramirez & Vamvakousis, 2012;

Rozgić et al., 2013; Womelsdorf & Everling, 2015. Of those studies, the most important is our own previous paper, the one that directly motivated this one, and the one that used an overlapping set of patients, but with different analyses and hypotheses: Xiao et al., 2023.

In this scan of the literature, we found that another typical range of beta is 12.5-30 Hz, as reported by Newson & Thiagarajan, 2019 after reviewing 184 EEG studies. We therefore repeated our analyses using this and observed similar results (specifically, with a $p = 0.0021$ during the delay period and $p < 10^{-4}$ during the feedback period).

Beta activity is subdivided into subbands in some previous studies, with varying outcomes observed among these subbands (Alagapan et al., 2023; Malekmohammadi et al., 2023; Rangaswamy et al., 2002). For example, only the low beta band activity demonstrates an initial acute decrease after deep brain stimulation for the treatment of depression followed by a subsequent increase (Alagapan et al., 2023). To further explore different components of beta activity, we also added post-hoc analysis and found that our results are mostly driven by lower beta. Below is the new figure (Supplementary Fig. 1) along with its caption, as well as the text describing these additional analyses.

To further confirm the result, we performed additional analysis using 12.5-30 Hz as the frequency range for beta activity (Newson & Thiagarajan, 2019). As in the original result, we found that beta activity showed a discernible difference between the rich and lean stimuli within the high response bias blocks (**Supplementary Fig. 1a**, $t(\text{stimulus}) = 3.1$, $p = 0.0021$, $\text{coef} = 5.1$, 95% confidence interval (95% CI) = [1.8 - 8.3], while no such distinction within the low response bias blocks ($t(\text{stimulus}) = -0.13$, $p = 0.90$, $\text{coef} = -0.22$, 95% confidence interval (95% CI) = [-3.6 - 3.2]). During the feedback period, beta activity during reward feedback was significantly larger than neutral feedback (**Supplementary Fig. 1b**, $t(\text{feedback}) = 12.9$, $p < 10^{-4}$, $\text{coef} = 17.6$, 95% confidence interval (95% CI) = [14.0 - 21.1]). We used time-frequency maps to investigate how various components of beta activity contribute to the overall effect (**Supplementary Fig. 1c-d**). Our findings indicate a large effect in the lower range of beta during both the delay and feedback periods, suggesting that our results are predominantly influenced by lower beta.

Supplementary Fig. 1: Additional analysis for beta activity. (a) Differences in beta power between the rich and lean trials during the delay period. (b) Differences in beta power between the reward and neutral trials during the feedback period. (c) Time-frequency plot for the difference comparing rich and lean stimulus during the delay period. Red indicates larger power towards rich stimulus. (d) Time-frequency plot for the difference comparing reward and neutral feedback during the feedback period. Red indicates a larger power towards reward feedback.

Minor points:

The Methods section states that none of the epilepsy patients had a mood disorder. How was this objectively assessed?

Thank you. To clarify, we mean instead that none of the epilepsy patients have been clinically diagnosed with depression. It is possible that these patients may have symptoms of a mood disorder but are not as severe as the Depression Cohort. We reworded the last paragraph of the introduction and incorporated this clarification into the Methods section to avoid potential confusion.

We recorded intracranial local field potentials (LFPs) from four reward-related regions in human participants performing the probabilistic reward task. In subjects without clinically diagnosed depression (“Epilepsy Cohort”, i.e., epilepsy patients undergoing intracranial seizure

monitoring). In subjects with medically refractory epilepsy undergoing intracranial seizure monitoring (“Epilepsy Cohort”, no clinical diagnosis of major depressive disorder), we find that enhanced beta (12-30 Hz) oscillations after decision choice in the ACC predict stronger biasing and also track reward receipt. On the other hand, in subjects with severe treatment-resistant depression (“Depression Cohort”, i.e., a cohort undergoing intracranial monitoring as part of a clinical trial studying depression; NCT03437928), both the behavioral bias and the neural response towards rewarding stimulus are reduced.

Participants

Fifteen participants (eight males and seven females, mean age 39 years, range 19-60 years) undergoing invasive monitoring for the treatment of refractory epilepsy at Baylor St. Luke’s Medical Center (Houston, Texas, USA) participated in our study. These participants did not carry a diagnosis of major depressive disorder. Implantation sites were determined solely by the clinical team for localization of the seizure onset zone. The Institution Review Board at Baylor College of Medicine approved this study (IRB protocol number H-18112), and all participants provided verbal and written consent to participate.

For p-value reporting, why not put the actual p-values? For example, line 108, line 110... They do this successfully in other parts of the paper, like line 136 and line 145.

We changed the original statement ‘ $p < 0.05$ for all three blocks’ to show the exact p value in each block.

Epilepsy Cohort participants showed a response bias: they tended to choose the more frequently rewarded stimulus more often than the less frequently rewarded one. Response bias averaged across all participants was larger than zero for all three blocks (Fig. 1b, one-sample t test compared with zero, block1: $p = 0.036$, block2: $p = 0.0029$, block3: $p = 0.0061$). Consistently, the accuracy for the rich stimulus was higher than the accuracy for the lean stimulus (Fig. 1c, paired-sample t-test, block1: $p = 0.026$, block2: $p = 0.0034$, block3: $p = 0.0086$).

Reviewer #3

Summary

This is well well-written and valuable manuscript investigating the neurophysiological correlates of reward choices in a unique data set of patients with major depressive disorder and a control group of epilepsy patients. The authors measured intracranial EEG (ACC, mOFC, amygdala) associated with a stochastically rewarded perceptual decision task in both patient groups. The behavioral data analysis demonstrated that responses to highly rewarding stimuli were more accurate compared to responses to less rewarding stimuli (this suggests a response bias) in the epilepsy patients. This response bias was blunted in the depression group. The epilepsy patients also showed increased reward anticipation in the delay period, as evidenced by increased beta activity in the ACC with rich compared to lean stimuli. This was additionally correlated with the enhanced response bias. Likewise, the receipt of rewarding feedback was linked with an increase in beta. The depression group did not demonstrate the reward-related increase in beta activity in

the ACC, neither during the delay period nor with the reward feedback. Both findings suggest a blunted reward response in the depression group.

We appreciate these positive comments and valuable summary.

My questions and comments are mainly regarding the (choices for specific) analyses of the data.

Data analysis and interpretation:

1. Did the authors analyze feedback-related activity in other frequency bands than beta and if so, what were the results for each group? If these were all null findings, please provide this in a footnote or supplementary section. Similarly, what were the findings for frequencies other than beta during the delay period in the depression group (were these null findings similar to the epilepsy group)?

We added the analysis on the delay-period activity and the feedback-period activity in other frequency bands than beta for each group. Below is the table we provided in the supplementary section, as well as the text describing these additional analyses.

When the trials were time-locked to the onset of feedback, we found that the reward response in depression patients was both reduced and delayed compared to the Epilepsy Cohort (Fig. 5f). These findings imply alterations in reward processing within the ACC among individuals with depression. Apart from the original analysis using beta activity, we also reported the result in other frequency bands for both the Epilepsy Cohort and the Depression Cohort (Supplementary Table. 2).

group	period	freq	coef.	std.err.	t	p	lower bound	upper bound
epilepsy	delay	delta	0.73	8.09	0.09	1	-15.12	16.57
epilepsy	delay	theta	-8.35	4.78	-1.75	0.40	-17.72	1.02
epilepsy	delay	alpha	-2.46	3.55	-0.69	1	-9.43	4.5
epilepsy	delay	gamma	-0.13	1.03	-0.12	1	-2.15	1.9
epilepsy	delay	highgamma	0.49	0.5	0.98	1	-0.49	1.48
epilepsy	feedback	delta	5.13	7.21	0.71	1	-9	19.27
epilepsy	feedback	theta	7.72	4.5	1.72	0.43	-1.1	16.53
epilepsy	feedback	alpha	15.52	3.36	4.62	<0.001	8.93	22.1
epilepsy	feedback	gamma	1.55	0.94	1.66	0.48	-0.28	3.39
epilepsy	feedback	highgamma	0.97	0.45	2.17	0.15	0.09	1.85
depression	delay	delta	4.59	3.56	1.29	0.99	-2.39	11.57
depression	delay	theta	2.82	2.52	1.12	1	-2.12	7.76
depression	delay	alpha	5.02	3.8	1.32	0.94	-2.44	12.47
depression	delay	gamma	-0.97	0.78	-1.24	1	-2.51	0.57
depression	delay	highgamma	-0.24	0.37	-0.64	1	-0.97	0.49
depression	feedback	delta	25.2	3.08	8.18	<0.001	19.16	31.23
depression	feedback	theta	7.97	2.85	2.8	0.03	2.39	13.56

depression	feedback	alpha	9.51	3.62	2.63	0.04	2.42	16.61
depression	feedback	gamma	2.39	0.68	3.52	0.002	1.06	3.72
depression	feedback	highgamma	0.51	0.32	1.58	0.57	-0.12	1.15

Bold denotes statistical significance at the $p < 0.05$ level.

If other frequencies were not analyzed, please provide an explanation for the focus on the beta band in the introduction, especially since the reward learning literature has frequently demonstrated the role of theta band activity in the ACC during reward-based learning, i.e. prediction errors and reward positivity (see for example Cavanagh & Frank, 2014; Lin et al. 2018)

We thank the reviewer for this suggestion. As noted by the reviewer, midfrontal theta increase usually reflects a need for increased cognitive control elicited by novel information, conflict, error, and negative feedback (Cavanagh & Frank, 2014; Lin et al., 2018). In experiments that include both monetary gains and losses, it has been shown that an increase in theta power was associated with losses while an increase in beta power was associated with gains (Cohen et al., 2007; HajiHosseini & Holroyd, 2015; Marco-Pallares et al., 2008). Thus, we chose to focus on the beta band in our study to investigate the effect of positive feedback. We added one paragraph in the introduction as an explanation for investigating beta activity. Though we now add the other frequencies (for completeness), we agree with the reviewer that our justification for focusing on the beta band could be stronger. We have now added the following new text to the Introduction and Discussion:

Several human electrophysiological studies have identified a mediofrontal oscillatory component associated with positive feedback in both gambling task and reversal learning task, tasks that have key features in common with our bias task. The increase observed in these tasks is in the beta range and occurs 200 to 400 ms after the feedback informing the participant about the monetary gains (Cohen et al., 2007; HajiHosseini et al., 2012; HajiHosseini & Holroyd, 2015; Marco-Pallares et al., 2008; Marco-Pallarés et al., 2015). It has been proposed that this beta activity is generated in the prefrontal cortex; the most commonly inferred source site is the dACC. It is further assumed that the ACC then transmits a fast motivational value signal, still in the beta band, from the frontal cortex to downstream reward-related regions (Marco-Pallarés et al., 2015). Moreover, beta activity in the cingulate cortex is of particular importance in depression (Clark et al., 2016; Huebl et al., 2016; Merkl et al., 2016). For example, a recent study showed that beta activity best tracks depressive states, seen as a decrease in beta band power during the first month of chronic stimulation, followed by an eventual rise (Alagapan et al., 2023). This result suggests that sustained, antidepressant responses might involve increased beta band power after prolonged stimulation.

Our study reveals that beta oscillations in the ACC, which represents reward outcome, are also elicited by rich stimuli and are correlated with behavioral preference. These findings provide further support for the importance of the ACC in reward learning. Indeed, these results are consistent with previous theories linking the ACC with cognition related to reward in general and to reward-mediated learning specifically. In experiments involving both monetary gains and losses, research has demonstrated that an increase in theta power is associated with losses, while

an increase in beta power is associated with gains (Cohen et al., 2007; HajiHosseini & Holroyd, 2015; Marco-Pallares et al., 2008). Aligned with prior findings, our study showed a rise in beta activity within the ACC following positive feedback. Future studies could investigate ACC activity during a similar task that incorporates negative feedback for a more comprehensive understanding.

2. Regarding the prediction error: Was there a relation between delay period beta activity (anticipation for a reward with rich vs lean stimuli) and the actual feedback related activity (either in beta or another frequency)? That is, did the size of the delay period activity impact the potential feedback-related prediction error when expectations were violated and was this different between the groups? See for example Unemoto & Holroyd, (2017) for the relation between reward anticipation and individual differences in depression.

We thank the reviewer for bringing up this interesting question. In the study by Unemoto and Holroyd (2017), each of the five cues was linked to a reward probability of 100%, 75%, 50%, 25%, or 0% for one of the two potential responses while the alternative response to each cue consistently led to no-reward feedback. They then calculated the reward positivity, a component of the human event-related potential triggered by unexpected reward delivery, which is suggested to reflect the reward prediction error signal. Our task is not designed to test the prediction error. We tried to do a similar analysis by calculating the difference between neural activity during feedback after the lean stimulus (20% chance of reward, more unexpected) and neural activity during feedback after the rich stimulus (60% chance of reward) as an approximation of prediction error. However, we didn't observe a significant correlation between the delay period beta activity and the prediction error ($p = 0.17$). This failure to achieve significance, of course, is not probative; this is especially true given that our dataset is limited by the EMU environment and our experiment has a smaller range of values. As such we do not think that reporting this information would provide any value in the paper.

3. Could the authors show the behavioral data in the depression group, i.e. what was the accuracy on the task in the depression cohort (separate for rich and lean stimuli) and what was their response bias across blocks?

As the reviewer suggested, we have performed the analyses using the behavioral data in the depression group. Below we provide the new figure (Supplementary Fig. 2) and caption as well as text describing these additional analyses. These results indicated that individuals with diagnosed depression did not exhibit a preference for stimuli linked to higher reward probabilities. This null finding is consistent with the conclusion that depression patients have blunted behavioral response towards reward.

We analyzed behavioral performance and electrophysiological patterns in these depression patients. We did not observe any significant difference between the accuracy for the rich stimulus and the accuracy for the lean stimulus (Supplementary Fig. 2a, paired-sample t-test, block1: $p = 0.90$, block2: $p = 0.94$, block3: $p = 0.60$). Response bias in depression patients was not significantly different from zero for all three blocks (Supplementary Fig. 2b, one-sample t test compared with zero, block1: $p = 0.93$, block2: $p = 0.83$ block3: $p = 0.38$). Compared to the epilepsy group, we found that response bias was blunted in these patients, especially in the last

block of the task (**Fig. 5b**, two-sample t -test compared with epilepsy patients, $t = 2.3$, $p = 0.030$). This indicates that the behavioral preference towards more frequently rewarded stimuli was reduced in this severely depressed group of individuals.

Supplementary Fig. 2: Behavioral performance in depression patients. (a) Accuracy for rich and lean stimuli averaged across all Depression Cohort patients. (b) Response bias averaged across all depression patients. A response bias value close to zero indicates there is no preference for choosing the more frequently rewarded stimulus.

4. Related to the previous question; if there was a difference in accuracy between the groups, how did this impact the frequency of reward feedback between the group? A higher reward frequency could impact the anticipation of a positive outcome over time and the associated beta activity in the delay period. Did the authors check the relative amount of reward received in relation to the size of the beta response during the delay period for each group?

We apologize for any confusion caused. Similar to the earlier study using the same task (Pizzagalli et al., 2005), we provided extra chances for reward feedback until a predetermined total amount of received rewards was met. If a participant failed to correctly identify the stimulus in a trial where reward feedback was scheduled, the feedback was delayed until the subsequent accurate identification of the same stimulus type. This ensured a consistent frequency of reward feedback for all participants.

We have now clarified this in the methods section:

The probability of reward for rich stimulus is three times more frequent than for lean stimulus (the other kind of mouth). If a participant didn't correctly identify the stimulus in a trial where reward feedback was scheduled, the feedback was postponed until the next accurate identification of the same stimulus type. Therefore, the consistency of reward feedback frequency was maintained across all participants for each stimulus type. This approach is the same as that used in the earlier study with the same task (Pizzagalli et al., 2005). The likelihood of receiving a reward was set at either 60% or 20%, depending on whether the choice was categorized as the rich stimulus or the lean stimulus.

Fig 5c: the beta power in the depression group seems to be almost null for both stimulus types, which makes it difficult to detect a difference between conditions, or to compare this data with the epilepsy group. Is the absence of beta in general related to depression (regardless of condition/task)? Please explain or elaborate on this in the discussion.

We appreciate the chance to clarify this point. These beta power values were z-scored across trials, irrespective of the stimulus types within each contact. As a result, they represent relative values rather than absolute ones. Thus, these smaller values shown in the plot indicate a smaller difference between conditions, rather than suggesting the absence of beta activity in general. We explained this in the Discussion.

Our results, which include both correlations with the behavior in non-clinically-depressed individuals and reduction in reward response in individuals with severe depression, suggest that beta activity within the ACC may be a biomarker for anhedonia. Such a biomarker has many potential benefits, including the ability to improve diagnosis and symptom monitoring. Moreover, they present an appealing target for neuromodulatory trials, which could focus on altering ACC beta and thereby reducing anhedonia. In our study, the average beta power values were z-scored across trials, regardless of stimulus types within each contact. Therefore, the smaller values in depression patients indicate a lesser difference between conditions, not the absence of beta activity overall. Monitoring this neural feature in a more naturalistic environment is essential for comparison with healthy controls and crucial for the development of potential treatments. In comparison to traditional questionnaires, continuous and passive monitoring of this neural feature in patients requires less effort from them, offers greater objectivity, and facilitates timely intervention. Future work will need to determine the time course of changes in this potential biomarker relative to those of depressive symptoms.

Discussion:

Overall, the discussion would benefit from explaining the current findings in the context of the previous literature on the neural response to rewarding stimuli with depression, i.e. the reward-related beta increase in ACC versus theta in previous studies. Also, previous behavioral findings may be less consistent than what is discussed here (line 292, 293); there are several studies that did not find a behavioral difference in reward learning between healthy controls and depressive cohort. Please include a discussion of those findings (see for example Davide et al. 2019 or Rothkirch et al. 2017)

We thank the reviewer for pointing out the inconsistency in behavioral findings in previous literature. While many studies report a diminished ability in depression patients to adapt their behavior based on past rewards (Morris et al., 2015; Pizzagalli et al., 2005, 2008; Vrieze et al., 2013; Whitmer et al., 2012), other studies have failed to observe such performance differences between participants with and without depression (Chase et al., 2010; Gheza et al., 2019; Gradin et al., 2011; Rothkirch et al., 2017). These conflicting findings regarding behavioral responses during reinforcement learning in depression could stem from the heterogeneity of depression or its stage. This behavioral effect might only be seen in individuals with depression who exhibit symptoms of anhedonia. For example, in the study by Chase et al., 2010, the decrease in sensitivity to reinforcement was independent of the participant group but was specifically linked to the severity of anhedonia. Elevated level of anhedonia has been linked to increased illness severity and episode duration (Gabbay et al., 2015). In our clinical trial, the participants from our group with treatment-resistant depression had not responded to at least four adequate depression treatments, indicating a high level of depression severity. The observed difference in response

bias between groups may be related to the advanced stage and elevated severity of depression in our study.

For the neural response, theta band activity in the ACC plays an important role during reinforcement learning (Cavanagh et al., 2010; Cavanagh & Frank, 2014; Li et al., 2018). For example, one previous study showed that midfrontal theta activity is correlated with the unsigned prediction error and is stronger for negative than for positive feedback (Mas-Herrero & Marco-Pallarés, 2014). Other studies demonstrated beta power increase over a frontal scalp area in response to rewarding events during reinforcement learning, potentially serving a critical role in transmitting motivational signals (HajiHosseini & Holroyd, 2015; Marco-Pallarés et al., 2015; van de Vijver et al., 2011). We added text in both the introduction and discussion sections to provide more information on the previous literature.

We have revised the discussion as shown below:

Both the behavioral bias and the neural response to rewarding stimuli are diminished in patients with depression, highlighting the changes in reward processing within the ACC in depression. ~~The behavioral finding confirms earlier work showing reduced reward learning in depressed patients relative to control subjects.~~ Previous studies have shown inconsistent results in behavioral findings during reinforcement learning in depression, possibly arising from the heterogeneity and stage of depression (Chase et al., 2010; Gradin et al., 2011; Morris et al., 2015; Pizzagalli et al., 2005, 2008; Rothkirch et al., 2017; Vrieze et al., 2013; Whitmer et al., 2012). Given that increased anhedonia levels are associated with greater illness severity and longer episodes, the observed response bias difference in our study may be attributed to the advanced stage and high severity of treatment-resistant depression (Gabbay et al., 2015).

Our study reveals that beta oscillations in the ACC, which represents reward outcome, are also elicited by rich stimuli and are correlated with behavioral preference. These findings provide further support for the importance of the ACC in reward learning. Indeed, these results are consistent with previous theories linking the ACC with cognition related to reward in general and to reward-mediated learning specifically. In experiments involving both monetary gains and losses, research has demonstrated that an increase in theta power is associated with losses, while an increase in beta power is associated with gains (Cohen et al., 2007; HajiHosseini & Holroyd, 2015; Marco-Pallares et al., 2008). Aligned with prior findings, our study showed a rise in beta activity within the ACC following positive feedback. Future studies could investigate ACC activity during a similar task that incorporates negative feedback for a more comprehensive understanding.

1. Lin, H., Saunders, B., Hutcherson, C. A., & Inzlicht, M. (2018). Midfrontal theta and pupil dilation parametrically track sub-jjective conflict (but also surprise) during intertemporal choice. *NeuroImage*, 172, 838– 852.
2. Cavanagh, J. F., & Frank, M. J. (2014). Frontal theta as a mechanism for cognitive control. *Trends in Cognitive Sciences*, 18(8), 414– 421
3. Umemoto, A., & Holroyd, C. B. (2017). Neural mechanisms of re-ward processing associated with depression- related personality traits. *Clinical Neurophysiology*, 128(7), 1184– 1196.
4. Abnormal approach-related motivation but spared reinforcement learning in MDD: Evidence from fronto-midline Theta oscillations and frontal Alpha asymmetry. Davide Gheza , Jasmina

Bakic, Chris Baeken , Rudi De Raedt , Gilles Pourtois Cogn Affect Behav Neurosci . 2019 Jun;19(3):759-777. doi: 10.3758/s13415-019-00693-4.

5. Neural mechanisms of reinforcement learning in unmedicated patients with major depressive disorder Marcus Rothkirch Jonas Tonn , Stephan Köhler, Philipp Sterzer. Brain. 2017 Apr 1;140(4):1147-1157. doi: 10.1093/brain/awx025.

Reviewer #4

The study by Xiao et al report an analysis on the involvement of beta oscillations in the anterior cingulate cortex (ACC) in modulating reward biases within the context of a probabilistic reward task. The findings revealed an increase in ACC beta power subsequent to the selection of a more frequently rewarded stimulus, and this was found to be positively associated with individual variations in behavioral biasing. These outcomes are taken to suggest that ACC beta oscillations may serve as indicators of reward learning processes. The paper is not only well-structured and easily comprehensible but also establishes a compelling rationale linking ACC functionality to the realms of reward and depression. The results presented lend strong support to the stated conclusions, rendering this paper a valuable addition to the existing body of knowledge on the neural underpinnings of reward processing and anhedonia. Nevertheless, I do have some specific comments on the manuscript, which I hope will aid the authors to improve the quality of their reporting – as detailed below:

We appreciate these positive comments.

Major comments

1. The authors may want to provide a rationale for their choice of investigating the beta band and substantiate its relevance to their research question by referencing prior literature.

We thank the reviewer for this thoughtful suggestion. We added the following paragraph in the introduction to provide a rationale for our choice of investigating the beta activity.

Several human electrophysiological studies have identified a mediofrontal oscillatory component associated with positive feedback in both gambling task and reversal learning task, tasks that have key features in common with our bias task. The increase observed in these tasks is in the beta range and occurs 200 to 400 ms after the feedback informing the participant about the monetary gains (Cohen et al., 2007; HajiHosseini et al., 2012; HajiHosseini & Holroyd, 2015; Marco-Pallares et al., 2008; Marco-Pallarés et al., 2015). It has been proposed that this beta activity is generated in the prefrontal cortex; the most commonly inferred source site is the dACC. It is further assumed that the ACC then transmits a fast motivational value signal, still in the beta band, from the frontal cortex to downstream reward-related regions (Marco-Pallarés et al., 2015). Moreover, beta activity in the cingulate cortex is of particular importance in depression (Clark et al., 2016; Huebl et al., 2016; Merkl et al., 2016). For example, a recent study showed that beta activity best tracks depressive states, seen as a decrease in beta band power during the first month of chronic stimulation, followed by an eventual rise (Alagapan et

al., 2023). This result suggests that sustained, antidepressant responses might involve increased beta band power after prolonged stimulation.

2. In the case of an exploratory analysis, it is essential to transparently discuss null results, including t-values, 95% confidence intervals, and p-values for other frequency bands. It is recommended to report the R-squared value of the mixed-effect model to ensure the statistical validity of the reported results. Please also include information about the residual distribution of the mixed-effect model to confirm the model's appropriateness. For each statistical test, please provide details on the degrees of freedom and the lower and upper bounds of the 95% confidence intervals.

We added Supplementary Table. 2 to report the t-values, 95% confidence intervals, and p-values for other frequency bands. Statistical results for the original analysis were added to the text in the results section.

group	period	freq	coef.	std.err.	t	p	lower bound	upper bound
epilepsy	delay	delta	0.73	8.09	0.09	1	-15.12	16.57
epilepsy	delay	theta	-8.35	4.78	-1.75	0.40	-17.72	1.02
epilepsy	delay	alpha	-2.46	3.55	-0.69	1	-9.43	4.5
epilepsy	delay	gamma	-0.13	1.03	-0.12	1	-2.15	1.9
epilepsy	delay	highgamma	0.49	0.5	0.98	1	-0.49	1.48
epilepsy	feedback	delta	5.13	7.21	0.71	1	-9	19.27
epilepsy	feedback	theta	7.72	4.5	1.72	0.43	-1.1	16.53
epilepsy	feedback	alpha	15.52	3.36	4.62	<0.001	8.93	22.1
epilepsy	feedback	gamma	1.55	0.94	1.66	0.48	-0.28	3.39
epilepsy	feedback	highgamma	0.97	0.45	2.17	0.15	0.09	1.85
depression	delay	delta	4.59	3.56	1.29	0.99	-2.39	11.57
depression	delay	theta	2.82	2.52	1.12	1	-2.12	7.76
depression	delay	alpha	5.02	3.8	1.32	0.94	-2.44	12.47
depression	delay	gamma	-0.97	0.78	-1.24	1	-2.51	0.57
depression	delay	highgamma	-0.24	0.37	-0.64	1	-0.97	0.49
depression	feedback	delta	25.2	3.08	8.18	<0.001	19.16	31.23
depression	feedback	theta	7.97	2.85	2.8	0.03	2.39	13.56
depression	feedback	alpha	9.51	3.62	2.63	0.04	2.42	16.61
depression	feedback	gamma	2.39	0.68	3.52	0.002	1.06	3.72
depression	feedback	highgamma	0.51	0.32	1.58	0.57	-0.12	1.15

Bold denotes statistical significance at the $p < 0.05$ level.

3. Please specify the number of permutations used for the cluster-corrected analysis.

The number of permutations=1000 for all cluster-corrected analyses in this manuscript. We have added this information to our manuscript.

Specifically, we found that event-aligned activity increases immediately after choice; this rise was significantly larger following choice of the rich stimulus (**Fig. 2c**, cluster-based permutation test, the significant cluster begins at 244 msec and ends at 293 msec after the choice, **number of permutations=1000**).

In the high response bias condition, one significant cluster (cluster-based permutation test, from 234 msec to 322 msec, **number of permutations=1000**) was found, while no significant cluster was found in the low response bias condition (**Fig. 3b**).

4. Specify the number of patients with electrode coverage in each of the four recording sites for both the epileptic and depression cohorts. Given the small size of the depressed cohort (N=4), consider discussing the challenges of attributing differences solely to depression and explore individual-level effects.

The electrodes were designed to be implanted in depression-relevant regions in the depressed cohort. Thus, all four depression patients have coverage in all four recording sites. The table below shows the number of channels in each of the four recording sites and the number of patients with electrode coverage in these sites. We have added this table to the manuscript.

*We recorded from four regions in the Epilepsy Cohort: anterior cingulate cortex (ACC), medial orbitofrontal cortex (mOFC), lateral orbitofrontal cortex (lOFC), and amygdala (Fig. 1e, **Supplementary Table. 1**).*

	ACC	mOFC	lOFC	Amygdala
Epilepsy	65(13)	91(10)	98(10)	42(6)
Depression	36(4)	55(4)	49(4)	31(4)

The numbers outside the parentheses indicate the total number of channels in each of the recording sites. The numbers inside the parentheses indicate the number of patients with electrode coverage in these sites.

5. Disclose whether the depression cohort was using any antidepressant medication and how the results were or were not different across the two groups. If medication was used, conduct an analysis to investigate the relationship between ACC beta power and duration and medication dosage. Please also ensure that the attenuated effect in the depression cohort is specific to the beta band in the ACC.

Our clinical trial requires a stable antidepressant medication regimen for the month preceding surgery and no changes during the in-patient monitoring period. One patient used fluoxetine while another patient used bupropion and venlafaxine. The remaining two patients were not receiving any psychotropic medication. With only four depression patients and constancy of medication dosing throughout our study, the study lacks sufficient power to explore the relationship between ACC beta power and duration and medication dosage. For other frequency bands, we did not observe significantly smaller reward response in the depression cohort. We have added the information on the use of any antidepressant medication in the revised Methods.

Each patient was implanted with permanent deep brain stimulation leads for stimulation delivery as well as with temporary sEEG electrodes for neural recordings. In our study, two patients used antidepressant medication, while the remaining two patients did not receive any medication. The trial protocol requires patients to maintain a stable dose of medication for at least one month before surgery, and no alterations are made to their medication during the in-patient monitoring period.

6. Validate that the depression cohort is free of comorbidities with other disorders, establishing it as a pure major depressive disorder cohort.

According to the criteria for excluding patients, the depression cohort is free of comorbidities with other disorders that might influence the result such as schizophrenia, bipolar disorder, personality disorder, alcohol or substance use disorder, seizure disorder, and neuro-developmental disorder. A more detailed version of the exclusion criteria can be found here: <https://clinicaltrials.gov/study/NCT03437928#participation-criteria>. We have added the following text in the Methods.

Four patients with treatment-resistant depression (two males and two females, mean age 42 years, range 37-58 years) who were enrolled in an early feasibility trial (NCT03437928) also participated in our study. This trial of individualized deep brain stimulation (DBS) guided by intracranial recordings is funded by the NIH BRAIN Initiative (UH3 NS103549) and approved by the Institution Review Board at Baylor College of Medicine (IRB number H-43036). These individuals did not carry significant psychiatric comorbidities based on the trial's exclusion of schizophrenia, bipolar disorder, personality disorders, and neuro-developmental disorders, as these conditions may impact the study results. Additional details regarding the exclusion criteria can be found: <https://clinicaltrials.gov/study/NCT03437928#participation-criteria>.

7. While the authors suggest ACC beta as a biomarker, it's important to discuss the feasibility of translating these findings to non-invasive EEG/MEG methods.

Non-invasive EEG and MEG exhibit a comparatively lower signal-to-noise ratio as the signals are attenuated by the skull and scalp. Although intracranial EEG offers more precise information due to direct contact with brain tissue, advanced source localization techniques can be employed in non-invasive methods to improve spatial accuracy. Future studies may aim to validate and replicate our observation of ACC beta activity in this task using non-invasive approaches. With ongoing advancements in experimental and analytical methods for EEG/MEG, we believe that this biomarker can be applied to a broader patient population in the future. We have added the text below to discuss this point.

Therefore, this biomarker holds the potential to offer clinicians a valuable temporally dynamic signal about the individual's ongoing state and the transitions they experience. Such a signal could alert clinicians to be watchful and influence decisions regarding potential therapeutic maneuvers such as medication adjustments, behavioral interventions, or modifications in stimulation delivery. Further investigation is needed to assess its effectiveness and practical application. Current technology enables the monitoring of beta activity at the stimulation site in freely moving Parkinsonian patients, allowing for the precise control of stimulation delivery

(Cagnan et al., 2019; Little et al., 2013; Rosa et al., 2015). Similar to this approach, it is possible to track the instantaneous power of the beta band in depression patients by deep brain stimulation systems with sensing capabilities. While the intracranial signal provides more precise anatomical information via direct contact with brain tissue, future studies should aim to validate and replicate our observation of ACC beta activity using non-invasive approaches. Wireless EEG headsets may be necessary to achieve more frequent and convenient measurements, while the implementation of advanced source localization techniques can enhance anatomical precision. Advancements in these areas can facilitate broader application across patient populations.

8. Clarify how you ruled out the possibility that the observed increase in beta values during both anticipation and receipt of rewards was not influenced by general decision-related processing or cognitive effort.

Thank you for bringing this to our attention. In our study, all neural activity was analyzed post the decision-making process and cognitive effort. We apologize for any confusion caused by not clearly specifying that the neural activity we analyzed occurs during either the delay period or the feedback period. To enhance clarity, we have modified the text in the results section to provide a more accurate and transparent description.

We first analyzed the delay period (Fig. 2a), which extends 500 msec after choice. The neural activity during this period occurs after the decision-making process. Cognitive effort typically occurs during the decision-making process, as individuals engage in mental processes such as evaluating options, weighing consequences, and selecting an action. Therefore, we do not think that the neural activity difference between different options during the delay period, after the participant has already made the choice, should be strongly influenced by cognitive effort.

9. Lastly, pls consider commenting on the precise anatomical location of the effects (within ACC) and comment on prior electrical stimulation studies linking dACC and motivation/drive (e.g., Parvizi, Neuron 2013 and Fox et al Nature Human Behavior 2020)

Prior research indicates that neighboring brain regions may serve distinct functions. For example, in Parvizi's study (Neuron, 2013), electrical stimulation of the anterior midcingulate cortex induces a 'will to persevere,' while stimulating subgenual or retrosplenial cingulate regions does not elicit any perceptual or behavioral responses. The channels in our study are mostly within the dorsal anterior cingulate cortex, with no channels in subgenual cingulate regions. Further studies with more recording sites in the ACC are needed to elucidate the roles of the different ACC subregions in reward processing. These studies with precise anatomical information may help with finer anatomical targeting to enhance therapeutic responses in future clinical trials. Fox et al., 2020 and Parvizi et al., 2013 have linked dACC with motivation and drive. The increase in ACC activity we observed in our study could represent an anticipation of reward, which may potentially translate into increased motivation to obtain the reward. We added the following text in the discussion to comment on this topic.

These results also have implications for our understanding of the role of the ACC. The ACC is thought to play a crucial role in reward processing. In non-human primates, individual ACC neurons process both experienced and fictive rewards to dynamically guide changes in behavior. Monkeys with ACC lesions are impaired in using rewarded trials to sustain the selection of the

correct object, emphasizing the importance of the ACC in reward-based decision-making. Our study reveals that beta oscillations in the ACC, which represents reward outcome, are also elicited by rich stimuli and are correlated with behavioral preference. Previous electrical stimulation studies suggest that dACC plays a crucial role in motivation and drive (Fox et al., 2020; Parvizi et al., 2013). In our study, we observed an increase in ACC activity when participants anticipated a reward. This anticipation of reward could potentially translate into an increased willingness to persevere and exert effort to obtain the reward. These stimulation studies also imply that adjacent brain regions may fulfill unique roles. Electrical stimulation of the anterior midcingulate cortex induces a 'will to persevere,' whereas stimulating subgenual or retrosplenial cingulate regions fails to evoke perceptual or behavioral responses. Further investigations with increased recording sites in the ACC are necessary to clarify the roles of its various subregions in reward processing.

References:

- Alagapan, S., Choi, K. S., Heisig, S., Riva-Posse, P., Crowell, A., Tiruvadi, V., Obatusin, M., Veerakumar, A., Waters, A. C., Gross, R. E., Quinn, S., Denison, L., O'Shaughnessy, M., Connor, M., Canal, G., Cha, J., Hershenberg, R., Nauvel, T., Isbaine, F., ... Rozell, C. J. (2023). Cingulate dynamics track depression recovery with deep brain stimulation. *Nature*, 622(7981), Article 7981. <https://doi.org/10.1038/s41586-023-06541-3>
- Babapoor-Farrokhran, S., Vinck, M., Womelsdorf, T., & Everling, S. (2017). Theta and beta synchrony coordinate frontal eye fields and anterior cingulate cortex during sensorimotor mapping. *Nature Communications*, 8(1), Article 1. <https://doi.org/10.1038/ncomms13967>
- Cagnan, H., Denison, T., McIntyre, C., & Brown, P. (2019). Emerging technologies for improved deep brain stimulation. *Nature Biotechnology*, 37(9), Article 9. <https://doi.org/10.1038/s41587-019-0244-6>
- Cavanagh, J. F., & Frank, M. J. (2014). Frontal theta as a mechanism for cognitive control. *Trends in Cognitive Sciences*, 18(8), 414–421. <https://doi.org/10.1016/j.tics.2014.04.012>
- Cavanagh, J. F., Frank, M. J., Klein, T. J., & Allen, J. J. B. (2010). Frontal Theta Links Prediction Errors to Behavioral Adaptation in Reinforcement Learning. *NeuroImage*, 49(4), 3198. <https://doi.org/10.1016/j.neuroimage.2009.11.080>
- Chase, H. W., Frank, M. J., Michael, A., Bullmore, E. T., Sahakian, B. J., & Robbins, T. W. (2010). Approach and avoidance learning in patients with major depression and healthy controls: Relation to anhedonia. *Psychological Medicine*, 40(3), 433–440. <https://doi.org/10.1017/S0033291709990468>
- Clark, D. L., Brown, E. C., Ramasubbu, R., & Kiss, Z. H. T. (2016). Intrinsic Local Beta Oscillations in the Subgenual Cingulate Relate to Depressive Symptoms in Treatment-Resistant Depression. *Biological Psychiatry*, 80(11), e93–e94. <https://doi.org/10.1016/j.biopsych.2016.02.032>
- Cohen, M. X., Elger, C. E., & Ranganath, C. (2007). Reward Expectation Modulates Feedback-Related Negativity and EEG Spectra. *NeuroImage*, 35(2), 968–978. <https://doi.org/10.1016/j.neuroimage.2006.11.056>

- Cooper, J. A., Arulpragasam, A. R., & Treadway, M. T. (2018). Anhedonia in depression: Biological mechanisms and computational models. *Current Opinion in Behavioral Sciences*, 22, 128–135. <https://doi.org/10.1016/j.cobeha.2018.01.024>
- Das, A., de los Angeles, C., & Menon, V. (2022). Electrophysiological foundations of the human default-mode network revealed by intracranial-EEG recordings during resting-state and cognition. *NeuroImage*, 250, 118927. <https://doi.org/10.1016/j.neuroimage.2022.118927>
- Der-Avakian, A., D'Souza, M. S., Pizzagalli, D. A., & Markou, A. (2013). Assessment of reward responsiveness in the response bias probabilistic reward task in rats: Implications for cross-species translational research. *Translational Psychiatry*, 3(8), e297. <https://doi.org/10.1038/tp.2013.74>
- Fox, K. C. R., Shi, L., Baek, S., Raccach, O., Foster, B. L., Saha, S., Margulies, D. S., Kucyi, A., & Parvizi, J. (2020). Intrinsic network architecture predicts the effects elicited by intracranial electrical stimulation of the human brain. *Nature Human Behaviour*, 4(10), Article 10. <https://doi.org/10.1038/s41562-020-0910-1>
- Gabbay, V., Johnson, A. R., Alonso, C. M., Evans, L. K., Babb, J. S., & Klein, R. G. (2015). Anhedonia, but not irritability, is associated with illness severity outcomes in adolescent major depression. *Journal of Child and Adolescent Psychopharmacology*, 25(3), 194–200. <https://doi.org/10.1089/cap.2014.0105>
- Gheza, D., Bakic, J., Baeken, C., De Raedt, R., & Pourtois, G. (2019). Abnormal approach-related motivation but spared reinforcement learning in MDD: Evidence from fronto-midline Theta oscillations and frontal Alpha asymmetry. *Cognitive, Affective, & Behavioral Neuroscience*, 19(3), 759–777. <https://doi.org/10.3758/s13415-019-00693-4>
- Gradin, V. B., Kumar, P., Waiter, G., Ahearn, T., Stickle, C., Milders, M., Reid, I., Hall, J., & Steele, J. D. (2011). Expected value and prediction error abnormalities in depression and schizophrenia. *Brain: A Journal of Neurology*, 134(Pt 6), 1751–1764. <https://doi.org/10.1093/brain/awr059>
- HajiHosseini, A., & Holroyd, C. B. (2015). Sensitivity of frontal beta oscillations to reward valence but not probability. *Neuroscience Letters*, 602, 99–103. <https://doi.org/10.1016/j.neulet.2015.06.054>
- HajiHosseini, A., Rodríguez-Fornells, A., & Marco-Pallarés, J. (2012). The role of beta-gamma oscillations in unexpected rewards processing. *NeuroImage*, 60(3), 1678–1685. <https://doi.org/10.1016/j.neuroimage.2012.01.125>
- Huebl, J., Brücke, C., Merkl, A., Bajbouj, M., Schneider, G.-H., & Kühn, A. A. (2016). Processing of emotional stimuli is reflected by modulations of beta band activity in the subgenual anterior cingulate cortex in patients with treatment resistant depression. *Social Cognitive and Affective Neuroscience*, 11(8), 1290–1298. <https://doi.org/10.1093/scan/nsw038>
- Insel, T., Cuthbert, B., Garvey, M., Heinssen, R., Pine, D. S., Quinn, K., Sanislow, C., & Wang, P. (2010). Research Domain Criteria (RDoC): Toward a New Classification Framework for Research on Mental Disorders. *American Journal of Psychiatry*, 167(7), 748–751. <https://doi.org/10.1176/appi.ajp.2010.09091379>
- Iturra-Mena, A. M., Kangas, B. D., Luc, O. T., Potter, D., & Pizzagalli, D. A. (2023). Electrophysiological signatures of reward learning in the rodent touchscreen-based Probabilistic Reward Task. *Neuropsychopharmacology*, 48(4), Article 4. <https://doi.org/10.1038/s41386-023-01532-4>

- Kangas, B. D., Wooldridge, L. M., Luc, O. T., Bergman, J., & Pizzagalli, D. A. (2020). Empirical validation of a touchscreen probabilistic reward task in rats. *Translational Psychiatry*, *10*(1), Article 1. <https://doi.org/10.1038/s41398-020-00969-1>
- Li, P., Peng, W., Li, H., & Holroyd, C. B. (2018). Electrophysiological measures reveal the role of anterior cingulate cortex in learning from unreliable feedback. *Cognitive, Affective, & Behavioral Neuroscience*, *18*(5), 949–963. <https://doi.org/10.3758/s13415-018-0615-3>
- Lin, H., Saunders, B., Hutcherson, C. A., & Inzlicht, M. (2018). Midfrontal theta and pupil dilation parametrically track subjective conflict (but also surprise) during intertemporal choice. *NeuroImage*, *172*, 838–852. <https://doi.org/10.1016/j.neuroimage.2017.10.055>
- Little, S., Pogosyan, A., Neal, S., Zavala, B., Zrinzo, L., Hariz, M., Foltynie, T., Limousin, P., Ashkan, K., FitzGerald, J., Green, A. L., Aziz, T. Z., & Brown, P. (2013). Adaptive Deep Brain Stimulation In Advanced Parkinson Disease. *Annals of Neurology*, *74*(3), 449–457. <https://doi.org/10.1002/ana.23951>
- Malekmohammadi, A., Ehrlich, S. K., Rauschecker, J. P., & Cheng, G. (2023). Listening to familiar music induces continuous inhibition of alpha and low-beta power. *Journal of Neurophysiology*, *129*(6), 1344–1358. <https://doi.org/10.1152/jn.00269.2022>
- Marco-Pallares, J., Cucurell, D., Cunillera, T., García, R., Andrés-Pueyo, A., Münte, T. F., & Rodríguez-Fornells, A. (2008). Human oscillatory activity associated to reward processing in a gambling task. *Neuropsychologia*, *46*(1), 241–248. <https://doi.org/10.1016/j.neuropsychologia.2007.07.016>
- Marco-Pallarés, J., Münte, T. F., & Rodríguez-Fornells, A. (2015). The role of high-frequency oscillatory activity in reward processing and learning. *Neuroscience & Biobehavioral Reviews*, *49*, 1–7. <https://doi.org/10.1016/j.neubiorev.2014.11.014>
- Mas-Herrero, E., & Marco-Pallarés, J. (2014). Frontal Theta Oscillatory Activity Is a Common Mechanism for the Computation of Unexpected Outcomes and Learning Rate. *Journal of Cognitive Neuroscience*, *26*(3), 447–458. https://doi.org/10.1162/jocn_a_00516
- Merkl, A., Neumann, W.-J., Huebl, J., Aust, S., Horn, A., Krauss, J. K., Dziobek, I., Kuhn, J., Schneider, G.-H., Bajbouj, M., & Kühn, A. A. (2016). Modulation of Beta-Band Activity in the Subgenual Anterior Cingulate Cortex during Emotional Empathy in Treatment-Resistant Depression. *Cerebral Cortex*, *26*(6), 2626–2638. <https://doi.org/10.1093/cercor/bhv100>
- Morris, B. H., Bylsma, L. M., Yaroslavsky, I., Kovacs, M., & Rottenberg, J. (2015). Reward Learning in Pediatric Depression and Anxiety: Preliminary Findings in a High-Risk Sample. *Depression and Anxiety*, *32*(5), 373–381. <https://doi.org/10.1002/da.22358>
- Newson, J. J., & Thiagarajan, T. C. (2019). EEG Frequency Bands in Psychiatric Disorders: A Review of Resting State Studies. *Frontiers in Human Neuroscience*, *12*. <https://www.frontiersin.org/articles/10.3389/fnhum.2018.00521>
- Parvizi, J., Rangarajan, V., Shirer, W. R., Desai, N., & Greicius, M. D. (2013). The Will to Persevere Induced by Electrical Stimulation of the Human Cingulate Gyrus. *Neuron*, *80*(6), 1359–1367. <https://doi.org/10.1016/j.neuron.2013.10.057>
- Pizzagalli, D. A., Iosifescu, D., Hallett, L. A., Ratner, K. G., & Fava, M. (2008). Reduced hedonic capacity in major depressive disorder: Evidence from a probabilistic reward task. *Journal of Psychiatric Research*, *43*(1), 76–87. <https://doi.org/10.1016/j.jpsychemes.2008.03.001>

- Pizzagalli, D. A., Jahn, A. L., & O'Shea, J. P. (2005). Toward an objective characterization of an anhedonic phenotype: A signal-detection approach. *Biological Psychiatry*, *57*(4), 319–327. <https://doi.org/10.1016/j.biopsych.2004.11.026>
- Ramirez, R., & Vamvakousis, Z. (2012). Detecting Emotion from EEG Signals Using the Emotive Epop Device. In F. M. Zanzotto, S. Tsumoto, N. Taatgen, & Y. Yao (Eds.), *Brain Informatics* (pp. 175–184). Springer. https://doi.org/10.1007/978-3-642-35139-6_17
- Rangaswamy, M., Porjesz, B., Chorlian, D. B., Wang, K., Jones, K. A., Bauer, L. O., Rohrbaugh, J., O'Connor, S. J., Kuperman, S., Reich, T., & Begleiter, H. (2002). Beta power in the EEG of alcoholics. *Biological Psychiatry*, *52*(8), 831–842. [https://doi.org/10.1016/s0006-3223\(02\)01362-8](https://doi.org/10.1016/s0006-3223(02)01362-8)
- Rosa, M., Arlotti, M., Ardolino, G., Cogiamanian, F., Marceglia, S., Di Fonzo, A., Cortese, F., Rampini, P. M., & Priori, A. (2015). Adaptive deep brain stimulation in a freely moving Parkinsonian patient. *Movement Disorders: Official Journal of the Movement Disorder Society*, *30*(7), 1003–1005. <https://doi.org/10.1002/mds.26241>
- Rothkirch, M., Tonn, J., Köhler, S., & Sterzer, P. (2017). Neural mechanisms of reinforcement learning in unmedicated patients with major depressive disorder. *Brain*, *140*(4), 1147–1157. <https://doi.org/10.1093/brain/awx025>
- Rozgić, V., Vitaladevuni, S. N., & Prasad, R. (2013). Robust EEG emotion classification using segment level decision fusion. *2013 IEEE International Conference on Acoustics, Speech and Signal Processing*, 1286–1290. <https://doi.org/10.1109/ICASSP.2013.6637858>
- Sailer, U., Wurm, F., & Pfabigan, D. M. (2023). Social and non-social feedback stimuli lead to comparable levels of reward learning and reward responsiveness in an online probabilistic reward task. *Behavior Research Methods*. <https://doi.org/10.3758/s13428-023-02255-6>
- van de Vijver, I., Ridderinkhof, K. R., & Cohen, M. X. (2011). Frontal Oscillatory Dynamics Predict Feedback Learning and Action Adjustment. *Journal of Cognitive Neuroscience*, *23*(12), 4106–4121. https://doi.org/10.1162/jocn_a_00110
- Vrieze, E., Pizzagalli, D. A., Demyttenaere, K., Hompes, T., Sienaert, P., de Boer, P., Schmidt, M., & Claes, S. (2013). Reduced reward learning predicts outcome in major depressive disorder. *Biological Psychiatry*, *73*(7), 639–645. <https://doi.org/10.1016/j.biopsych.2012.10.014>
- Wang, S., Leri, F., & Rizvi, S. J. (2021). Anhedonia as a central factor in depression: Neural mechanisms revealed from preclinical to clinical evidence. *Progress in Neuro-Psychopharmacology & Biological Psychiatry*, *110*, 110289. <https://doi.org/10.1016/j.pnpbp.2021.110289>
- Whitmer, A. J., Frank, M. J., & Gotlib, I. H. (2012). Sensitivity to reward and punishment in major depressive disorder: Effects of rumination and of single versus multiple experiences. *Cognition and Emotion*, *26*(8), 1475–1485. <https://doi.org/10.1080/02699931.2012.682973>
- Wilkinson, M. P., Slaney, C. L., Mellor, J. R., & Robinson, E. S. J. (2021). Investigation of reward learning and feedback sensitivity in non-clinical participants with a history of early life stress. *PLOS ONE*, *16*(12), e0260444. <https://doi.org/10.1371/journal.pone.0260444>

- Womelsdorf, T., & Everling, S. (2015). Long-Range Attention Networks: Circuit Motifs Underlying Endogenously Controlled Stimulus Selection. *Trends in Neurosciences*, 38(11), 682–700. <https://doi.org/10.1016/j.tins.2015.08.009>
- Xiao, J., Provenza, N. R., Asfour, J., Myers, J., Mathura, R. K., Metzger, B., Adkinson, J. A., Allawala, A. B., Pirtle, V., Oswald, D., Shofty, B., Robinson, M. E., Mathew, S. J., Goodman, W. K., Pouratian, N., Schrater, P. R., Patel, A. B., Tolia, A. S., Bijanki, K. R., ... Sheth, S. A. (2023). Decoding Depression Severity From Intracranial Neural Activity. *Biological Psychiatry*. <https://doi.org/10.1016/j.biopsych.2023.01.020>

REVIEWERS' COMMENTS

Reviewer #1 (Remarks to the Author):

Very nice revision and very detailed response to all reviewers. I have no further comments.

Reviewer #1 (Remarks on code availability):

N/A

Reviewer #2 (Remarks to the Author):

The reviewers did an absolutely wonderful job addressing my critiques.

Reviewer #3 (Remarks to the Author):

The authors have addressed all my concerns with great detail. Thank you

Reviewer #4 (Remarks to the Author):

Thanks for addressing my points. I have no further comments.